# Epigenomic diversity of cortical projection neurons in the mouse brain

Zhuzhu Zhang[1,13], Jingtian Zhou[1,2,13], Pengcheng Tan[1,3], Yan Pang[4], Angeline C. Rivkin[1], Megan A. Kirchgessner[4,5], Elora Williams[6], Cheng-Ta Lee[7], Hanqing Liu[1,8], Alexis D. Franklin[4], Paula Assakura Miyazaki[4], Anna Bartlett[1], Andrew I. Aldridge[1], Minh Vu[4], Lara Boggeman[9], Conor Fitzpatrick[9], Joseph R. Nery[1], Rosa G. Castanon[1], Mohammad Rashid[4], Matthew W. Jacobs[4], Tony Ito-Cole[4], Carolyn O'Connor[9], António Pinto-Duartec[10], Bertha Dominguez[7], Jared B. Smith[6], Sheng-Yong Niu[1], Kuo-Fen Lee[7], Xin Jin[6], Eran A. Mukamel[11], M. Margarita Behrens[10], Joseph R. Ecker[1,12 ✉] & Edward M. Callaway[4 ✉]

Neuronal cell types are classically defined by their molecular properties, anatomy and functions. Although recent advances in single-cell genomics have led to high-resolution molecular characterization of cell type diversity in the brain[1], neuronal cell types are often studied out of the context of their anatomical properties. To improve our understanding of the relationship between molecular and anatomical features that define cortical neurons, here we combined retrograde labelling with single-nucleus DNA methylation sequencing to link neural epigenomic properties to projections. We examined 11,827 single neocortical neurons from 63 cortico-cortical and cortico-subcortical long-distance projections. Our results showed unique epigenetic signatures of projection neurons that correspond to their laminar and regional location and projection patterns. On the basis of their epigenomes, intra-telencephalic cells that project to different cortical targets could be further distinguished, and some layer 5 neurons that project to extra-telencephalic targets (L5 ET) formed separate clusters that aligned with their axonal projections. Such separation varied between cortical areas, which suggests that there are area-specific differences in L5 ET subtypes, which were further validated by anatomical studies. Notably, a population of cortico-cortical projection neurons clustered with L5 ET rather than intra-telencephalic neurons, which suggests that a population of L5 ET cortical neurons projects to both targets. We verified the existence of these neurons by dual retrograde labelling and anterograde tracing of cortico-cortical projection neurons, which revealed axon terminals in extra-telencephalic targets including the thalamus, superior colliculus and pons. These findings highlight the power of single-cell epigenomic approaches to connect the molecular properties of neurons with their anatomical and projection properties.

The mammalian brain is a complex system that consists of several types of neuron with diverse morphology, physiology, connections, gene expression and epigenetic modifications. Identifying brain cell types and how they interact is crucial to understanding the neural mechanisms that underlie brain function. Single-cell technologies deconvolve mammalian brains into molecularly defined cell clusters that correspond to putative neuron types[1]. However, the correspondence between molecular cell types and neuronal populations defined by connectivity are largely unknown.

Previous single-cell analyses have revealed transcriptomic clusters and linked them to neuron types with different projection patterns in a few particular brain regions[2–5]. For the cerebral cortex, the most prominent molecular distinction related to projection targets is the separation of cortical neurons into distinct and apparently non-overlapping intra-telencephalic (IT) and L5 ET (also known as pyramidal tract) groups. In some cases, L5 ET cells have been further divided on the basis of both gene expression and corresponding axon projections[2]. Although the separation of L5 IT and ET neurons seems to be conserved

[1]Genomic Analysis Laboratory, The Salk Institute for Biological Studies, La Jolla, CA, USA. [2]Bioinformatics and Systems Biology Program, University of California San Diego, La Jolla, CA, USA. [3]School of Pharmaceutical Sciences, Tsinghua University, Beijing, China. [4]Systems Neurobiology Laboratories, The Salk Institute for Biological Studies, La Jolla, CA, USA. [5]Neurosciences Graduate Program, University of California, San Diego, La Jolla, CA, USA. [6]Molecular Neurobiology Laboratory, The Salk Institute for Biological Studies, La Jolla, CA, USA. [7]Peptide Biology Laboratories, The Salk Institute for Biological Studies, La Jolla, CA, USA. [8]Division of Biological Sciences, University of California San Diego, La Jolla, CA, USA. [9]Flow Cytometry Core Facility, The Salk Institute for Biological Studies, La Jolla, CA, USA. [10]Computational Neurobiology Laboratory, The Salk Institute for Biological Studies, La Jolla, CA, USA. [11]Department of Cognitive Science, University of California, San Diego, La Jolla, CA, USA. [12]Howard Hughes Medical Institute, The Salk Institute for Biological Studies, La Jolla, CA, USA. [13]These authors contributed equally: Zhuzhu Zhang, Jingtian Zhou. ✉e-mail: ecker@salk.edu; callaway@salk.edu

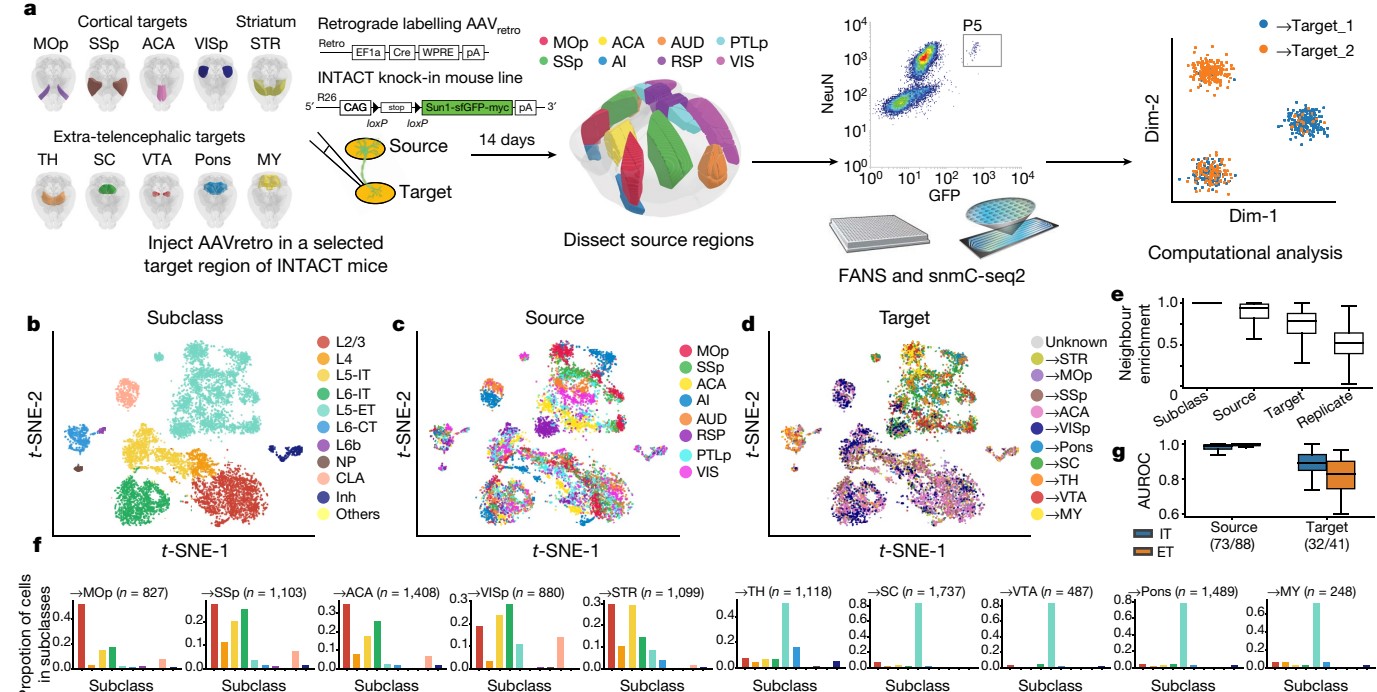

**Fig. 1 | The epigenomic landscape of cortical projection neurons.**
**a**, Schematics of the epi-retro-seq workflow. SC, superior colliculus; MY, medulla; STR, striatum; TH, thalamus. All brain atlas images were created based on Wang et al.[25] and ©2017 Allen Institute for Brain Science. Allen Brain Reference Atlas. Available from: http://atlas.brain-map.org.
**b–d**, Two-dimensional t-SNE of 11,827 cortical neuron nuclei on the basis of mCH levels in 100-kb genomic bins, coloured by subclass (**b**), the source of neurons (**c**), or their projection target (**d**). Inh, inhibitory; NP, near-projecting; CLA, claustrum. **e**, Neighbour enrichment scores of cells categorized by subclass (n = 11,827), source (n = 11,827), target (n = 10,396) and replicate (n = 11,638). **f**, The distribution across cell subclasses of neurons that projected to each IT (left) or ET (right) target. **g**, AUROC of distinguishing between source pairs or target pairs computed for IT and ET neurons on the basis of gene body mCH (n = 73, 88, 32 and 41; from left to right). For all box plots, centre line denotes the median; box limits denote first and third quartiles; and whiskers denote 1.5 × the interquartile range.

across cortical areas[6] and species[7], a systematic analysis of the relationships between a larger set of projection targets and molecular identities across several cortical areas has not been conducted. To what extent cortical projection neuron types can be further distinguished or divided by incorporating anatomical information with molecular analyses, and whether these cell types and correspondences are conserved across cortical areas, is unclear.

## Epi-retro-seq of 63 cortical projections

To address these questions, we developed epi-retro-seq, which applies single-nucleus methylome sequencing (snmC-seq)[8] to neurons dissected from cortical source regions that were labelled on the basis of their long-distance projections to specific cortical and subcortical targets (Fig. 1a). In epi-retro-seq, the retrograde viral tracer rAAV2-retro-Cre[9] is injected in the target region in an INTACT mouse[10], turning on Cre-dependent nuclear expression of green fluorescent protein (GFP) in neurons that project to the injected target, throughout the mouse brain. Source regions of interest were manually dissected (Methods), and GFP+NeuN+ nuclei (the GFP-labelled projection neurons) were isolated as single nuclei using fluorescence-activated nuclei sorting (FANS) and assayed using snmC-seq2[8]. snmC-seq enables the identification of potential regulatory elements and a prediction of gene expression in the same neurons[10–12]. In addition, methylation at non-CG (CH; in which 'H' denotes A, T or C) dinucleotides (mCH) accumulates, and methylation at CG dinucleotides (mCG) reconfigures during the development of cortical synapses, which suggests possible links between epigenetics and connectivity[13,14].

We performed epi-retro-seq to characterize projection neurons from 8 mouse cortical areas ('source') that project to 10 cortical or

subcortical regions ('target'), covering 26 cortico-cortical (CC) projections and 37 cortico-subcortical projections (Supplementary Table 1). The ten injected target regions include four cortical areas (the primary motor cortex (MOp), primary somatosensory cortex (SSp), anterior cingulate area (ACA), and primary visual cortex (VISp)), and six major subcortical structures (the striatum, thalamus, superior colliculus, ventral tegmental area (VTA) and substantia nigra, pons and medulla). The eight dissected source cortical regions are MOp, SSp, ACA, agranular insular cortex (AI), retrosplenial area (RSP), auditory cortex (AUD), posterior parietal cortex (PTLp) and visual cortex (VIS) (Extended Data Fig. 1).

## Methylome of cortical projection neurons

After quality control procedures (Methods), we obtained high-quality methylomes for 11,827 single cortical projection neurons (Extended Data Fig. 2). The mCH level in each single nucleus was computed across the genome using 100-kb genomic bins and used to perform unsupervised clustering of the projection neurons. Overall, the cortical projection neuron clusters were annotated into ten subclasses (Fig. 1b) on the basis of reduced levels of gene body mCH—a proxy for gene expression—of known marker genes (Methods). Results from cluster analyses and annotation were used to conduct a further quality check to identify neurons with projection targets that could not be confidently assigned owing to potential artefacts (Methods). We identified 1,431 neurons from experiments in which the projection target could not be confidently assigned (Extended Data Fig. 2i), leaving 10,396 neurons with confident projection target assignments. All subsequent analyses that incorporate projection target information are restricted to these neurons.

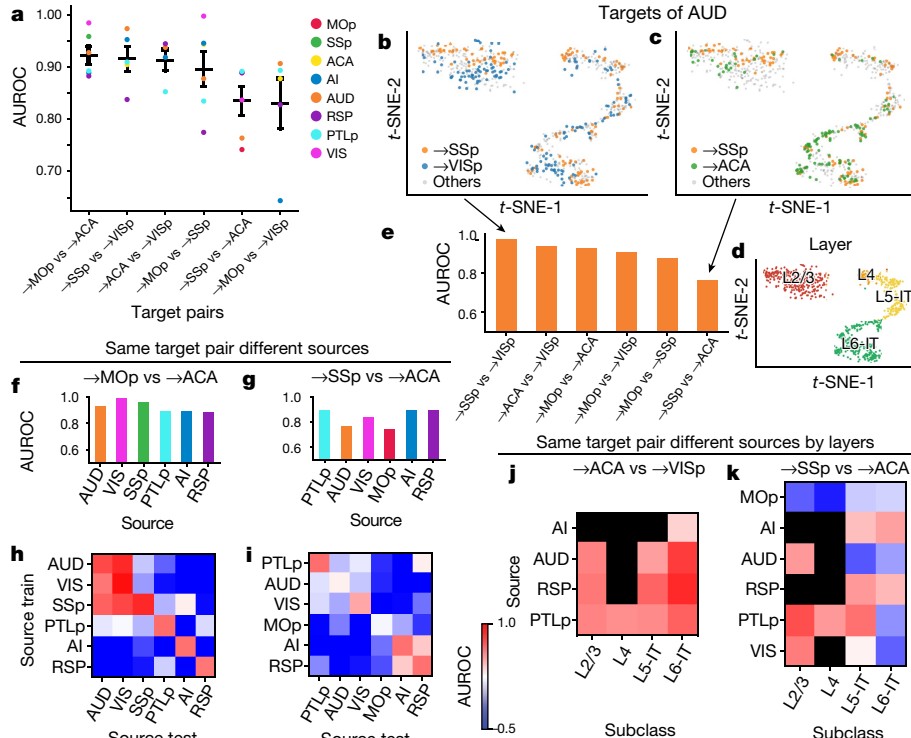

**Fig. 2 | Epigenetic differences between IT neurons projecting to different targets. a**, AUROC to distinguish cortical neurons projecting to one cortical target versus another. Data are mean ± s.e.m. (*n* = 6, 5, 4, 6, 6 and 5 sources; left to right). **b**–**d**, *t*-SNE of AUD neurons in the IT subclasses (*n* = 737) coloured by projections (**b**, **c**) and subclasses (**d**). **e**, AUROC to distinguish AUD neurons projecting to each target pair. **f**, **g**, The AUROC for comparisons between MOp and ACA-projecting neurons (**f**), and between SSp and ACA-projecting neurons (**g**) from different sources. **h**, **i**, Heat maps of AUROC from prediction models that were trained on one source (row) and tested on another source (column) to distinguish between neurons projecting to MOp and ACA (**h**), or between neurons projecting to SSp and ACA (**i**). **j**, **k**, Heat maps of AUROC from prediction models that were trained and tested on neurons from each cortical layer (column) in each source (row), to distinguish between ACA and VISp-projecting neurons (**j**), or between SSp and ACA-projecting neurons (**k**).

Within each cell subclass, excitatory but not inhibitory neurons from different cortical regions were further separated from each other (Fig. 1c), which demonstrates the distinct spatial DNA methylation patterns in cortical projection neurons. The cell subclasses and spatial patterns in epi-retro-seq were in agreement with those in snmC-seq data from the same cortical regions without enrichment of specific projections (Extended Data Fig. 3a). Neurons projecting to different target regions were more similar within each subclass than neurons from different source regions (Fig. 1d), indicating that they shared a more similar DNA methylation landscape. Neighbour enrichment scores were used to quantify the variations of DNA methylation that originated from different cell types, cortical spatial regions and projection targets (Methods). Neurons from the same subclass occupied highly similar regions in the dimension reduction space (neighbour enrichment score was close to 1) (Fig. 1e). Scores were also high for comparisons across neurons from the same source, followed by projections to the same target. Scores were near chance for biological replicates (neighbour enrichment score of 0.5), which indicates that the mCH profiles of different replicates are highly consistent (Fig. 1e).

Although neurons projecting to different target regions were not completely separated on the *t*-distributed stochastic neighbour embedding (*t*-SNE), we observed an explicit enrichment of CC and cortico-striatal projection neurons in IT subclasses (L2/3, L4, L5 IT, L6 IT and claustrum), separated from neurons that project to the remaining structures outside the telencephalon, which were categorized as L5 ET neurons (Fig. 1f, Extended Data Fig. 3). The enrichment is highly consistent across source regions (Extended Data Fig. 3b). As expected, many corticothalamic projection neurons were also found in the L6 corticothalamic subclass (Fig. 1f, Extended Data Fig. 3). These enrichment patterns are consistent with our knowledge about laminar enrichment of the projection neurons, which reflects the high quality of our retrogradely labelled single-nuclei methylation dataset.

To quantify methylation differences between neurons from different source regions or projecting to different target regions further, we used the area under the receiver operating characteristic curve (AUROC) of linear models trained to distinguish source pairs or target pairs on the basis of mCH (Methods). We found that most neurons dissected from different source regions could be well separated (Fig. 1g). Most of the neurons projecting to different target regions were also separable by mCH in this supervised setting (Fig. 1g), although they were closely mixed in the unsupervised embeddings (Fig. 1d). These findings indicate that nearly all of the different types of projection neuron that were profiled have differences in their epigenomes. Further analyses of these quantitative differences, described below, allowed the assessment of possible organizational principles that might exist in the relationships between DNA methylation, projections targets and sources, including both areal and laminar sources.

## Predicting IT neuron targets with mCH

In total, 42.6% of the cortical projection neurons profiled in our epi-retro-seq data were identified as IT neurons, and annotated according to their presumptive cortical layers (Fig. 1b). We investigated the contribution of the cortical area in which cell bodies were located versus their cortical projection targets, to the variation of their DNA methylation profiles. We focused on 26 CC projections from 8 cortical areas to 4 different cortical targets. All possible pairs of 4 cortical targets were assessed for each of the 8 sources to generate 32 AUROC scores, organized according to projection target pairs (Fig. 2a, Extended

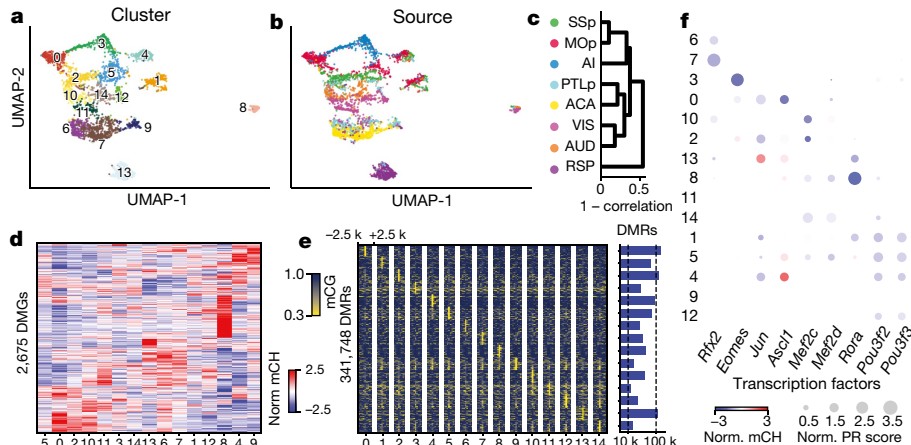

**Fig. 3 | Epigenetic diversity of L5 ET neurons. a**, **b**, Fifteen clusters of L5 ET neurons (*n* = 4,176) shown on the UMAP plot, coloured by cluster (**a**), or the source of neurons (**b**). **c**, Dendrogram shows the correlations between mCH profiles of L5 ET neurons from different sources. **d**, Gene body mCH levels in each cluster of 2,675 CH-DMGs that were identified in pairwise comparisons between L5 ET clusters. **e**, A total of 341,748 CG-DMRs were identified across the 15 L5 ET clusters. Left, the mCG levels at CG-DMRs and their 2.5-kb flanking genomic regions in each cluster were visualized in the heat map. Right, the numbers of CG-DMRs hypomethylated in each cluster were plotted in the bar chart. **f**, Examples of some predicted key regulator transcription factors. The size of each dot represents the normalized PageRank (PR) score of the transcription factor. The colour of the dot represents the gene body mCH of the transcription factor in the corresponding L5 ET cluster.

Data Fig. 4a–d). Among the six projection target pairs examined, neurons projecting to the MOp versus ACA were the most distinguishable (average AUROC value of 0.922), similar to neurons projecting to the SSp versus VISp and the ACA versus VISp (average AUROC values of 0.915 and 0.914, respectively), whereas neurons that project to the SSp versus ACA and to the MOp versus VISp were the least separable (average AUROC values of 0.837 and 0.831, respectively) (Fig. 2a). In addition, for each target pair, the performance of the predictive model varied among neurons from different source cortical regions (Fig. 2a, Extended Data Fig. 4a–d).

These analyses suggest that epigenetic differences between CC projection neurons depend on a combination of both the specific targets to which neurons project and the sources where the neurons reside. For example, among AUD IT neurons, AUD–SSp (projecting from AUD to SSp) neurons were better separated from AUD–VISp neurons (AUROC value of 0.974) (Fig. 2b, e) than from AUD–ACA neurons (AUROC value of 0.766) (Fig. 2c, e). The distinctions between these projections did not arise from different distributions across layers (Fig. 2d). This demonstrates that the level of epigenetic differences between AUD IT neurons varies depending on their projection targets. Similarly, when comparing neurons from different sources projecting to the same target pair, we observed different levels of distinguishability in our models. For example, although neurons projecting to the MOp versus neurons projecting to the ACA were more distinguishable (that is, had higher AUROC scores) than neurons projecting to the SSp versus those projecting to the ACA, we observed variation of the AUROC scores across different source regions for both target pairs (Fig. 2f, g). To further examine whether the same epigenetic differences that distinguished target pairs for one source might be conserved across sources, we trained models to predict targets using neurons from one source and then tested it on another source (Methods). Notably, these cross-source models can distinguish target pairs in many cases, whereas the performance of models trained on any particular region varied in their ability to predict projections from other regions (Fig. 2h, i, Extended Data Fig. 4e–h). For example, the model trained on AUD performed better in distinguishing VIS–MOp versus VIS–ACA neurons than the models trained on RSP or PTLp (Fig. 2h). This suggests that AUD and VIS neurons are more similar to each other in the molecular markers that distinguish neurons projecting to MOp versus ACA than other cortical areas. These results indicate that cortical regions might

form different groups with shared correlations between molecular markers and projection targets.

In addition, we assessed the level of distinguishability between two cortical targets, both for neurons within the same layer and for neurons in different layers (Fig. 2j, k, Extended Data Fig. 5a–c). By training and testing the predictive models in each layer separately, we typically observed higher distinguishability between ACA-projecting and VISp-projecting neurons than between SSp-projecting and ACA-projecting neurons (Fig. 2j, k). However, predictions for SSp-projecting and ACA-projecting neurons were more variable, with some sources being better than others for all layers (for example, MOp versus PTLp) (Fig. 2k) and some layers being better than others, even for the same source (for example, AUD and VIS) (Fig. 2k). We further tested whether cross-layer-trained models could distinguish the projection targets (Methods), and observed that the performance was generally comparable to within-layer models (Extended Data Fig. 5d–f). These results suggest that there may be shared epigenetic signatures across layers that contribute to correlations with the projection targets.

Furthermore, we identified differentially methylated genes (DMGs) at CH sites (CH-DMGs) between different pairs of CC projection neurons in each source region using hierarchical linear models. In total, 1,644 CH-DMGs were identified (Extended Data Fig. 5g, Supplementary Table 3), among which 1,497 (91.1% of CH-DMGs) were statistically significant in only one source region. The fact that most CH-DMGs were unique to one source region suggests that different genes may participate in defining projections from different source regions. Gene ontology (GO) enrichment analysis revealed that CH-DMGs were enriched for genes that participate in intracellular transport and the regulation of synapse structure (Supplementary Table 3), and might differ between neurons with different projections. For example, *Bsn* is differentially methylated between MOp-projecting and SSp-projecting neurons in the AUD and VIS (Extended Data Fig. 5g). It encodes a presynaptic cytomatrix scaffolding protein (bassoon) that is primarily expressed in neurons, and is essential for the regulation of neurotransmitter release[15]. *Scn2a1* encodes a voltage-dependent sodium channel protein (SCN2A1) and is differentially methylated between ACA-projecting and VISp-projecting neurons in the AI and PTLp (Extended Data Fig. 5g). This channel regulates neuronal excitability, and variants are associated with autism and seizure disorders[16].

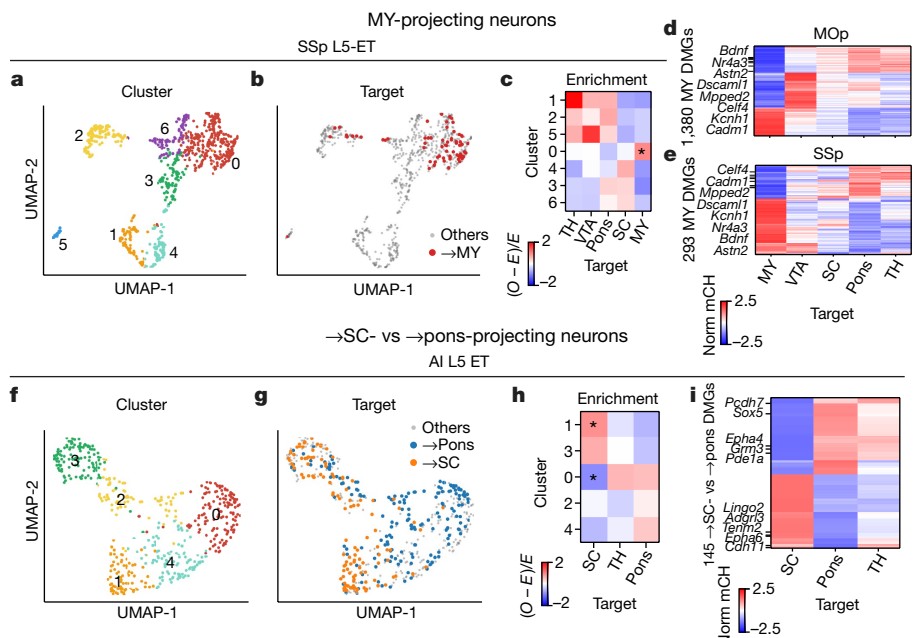

**Fig. 4 | Epigenetic differences between L5 ET neurons projecting to different targets.**
**a**, **b**, **f**, **g**, UMAP of SSp (**a**, $n = 884$) or AI (**f**, $n = 531$) L5 ET neurons by 100 kb-bin mCH are coloured by clusters (**a**, **f**) or projection targets (**b**, **g**). **c**, **h**, The enrichment of SSp (**c**) or AI (**h**) neurons projecting to each target in each cluster (asterisk represents FDR < 0.05). *E*, expected; *O*, observed. **d**, **e**, **i**, Gene body mCH levels of the CH-DMGs in the MOp (**d**) or SSp (**e**) between neurons projecting to the medulla and other ET targets, or in AI between neurons projecting to the superior colliculus and pons (**i**). Values are *Z*-score normalized by rows. Examples of CH-DMGs hypomethylated in both MOp–medulla and SSp–medulla neurons are labelled in **d** and **e**.

## Epigenetically distinct L5 ET subtypes

L5 ET neurons are the most abundant cell population in our datasets (4,176 (35.3%) single neurons), and are 6.3-fold enriched in epi-retro-seq compared to the total number of neurons observed in unbiased snmC-seq2 profiling. This provides us with a unique opportunity to investigate subpopulations of L5 ET neurons more closely. L5 ET neurons further segregated into 15 clusters (Fig. 3a). Much of the separation between clusters was driven by the source location of the neurons, as neurons from different sources were clearly separated on the UMAP (Fig. 3b), and each of the clusters consists of neurons mostly from one or two sources (Extended Data Fig. 6a). The similarities between L5 ET neurons from different sources (Fig. 3c) were not well explained by their spatial proximity anterior-posteriorly or medial-laterally, but better correlated with the anatomical and functional connectivity between these regions. For example, MOp and SSp are components of the somatic sensorimotor subnetwork, whereas AUD, VIS, ACA and PTLp are components of the medial subnetwork that channels information between sensory areas (that include VISp and AUD) and higher-order association areas (that include PTLp and ACA)[17].

To further explore the molecular identity of these L5 ET clusters, we identified 2,675 CH-DMGs (Fig. 3d, Extended Data Fig. 6c, Supplementary Table 4) and 341,748 CG-DMRs (Fig. 3e, Supplementary Table 5) that were hypomethylated in the corresponding L5 ET clusters. GO enrichment analysis revealed that these CH-DMGs were enriched in genes involved in cell communication, neurogenesis, cell morphogenesis and axon guidance (Supplementary Table 4). The average length of CG-DMRs was 227 base pairs (bp), and 84.9% of the CG-DMRs were distal elements that located more than 5 kb from the annotated transcription start sites. The level of mCH at gene bodies is inversely correlated with gene expression, whereas the level of mCG at gene regulatory elements, such as promoters and enhancers, is inversely correlated with their regulatory activities. These relationships allowed us to use a gene regulatory network-based method to integrate this information and identify transcription factors that might function as key regulators in each cluster (Methods, Fig. 3f, Extended Data Fig. 6d, e). For example, the transcriptional activator *Rora* was scored as one of the top transcription factors and is hypo-CH-methylated in clusters 1, 8 and 13, and especially in cluster 8, indicating its potential expression. The binding motif of RORA was also enriched in the CG-DMRs of these same clusters, which suggests that RORA may bind to *cis*-regulatory

elements that in turn regulate a set of predicted downstream target genes. Many of these target genes are related to brain functions and are also hypomethylated in cluster 8 (Extended Data Fig. 6f).

## L5 ET subtypes project differently

Neurons from the same sources (except AI and RSP) distributed into more than one cluster (Fig. 3a, b, Extended Data Fig. 6b), which prompted us to ask whether some of the differences between L5 ET clusters also correspond to the different projection targets. To investigate this, we performed another iteration of cluster analysis using L5 ET cell data from each of the source regions separately, and identified finer L5 ET clusters within each source region (Extended Data Fig. 7a).

Among all comparisons between projection targets and clusters, neurons projecting to the medulla were most distinct. SSp L5 ET neurons further segregated into seven clusters (Fig. 4a), among which SSp–medulla neurons showed a clear enrichment in cluster 0 (false discovery rate (FDR) = $3.69 \times 10^{-2}$, Wald test) (Fig. 4b, c). Similarly, we identified seven clusters of MOp L5 ET neurons, and MOp–medulla neurons were also significantly enriched in one of the clusters (FDR = $1.44 \times 10^{-2}$, Wald test) (Extended Data Fig. 7c, d). Moreover, neurons projecting to the medulla were robustly distinguished from other L5 ET neurons in our prediction models for both MOp and SSp (average AUROC scores of 0.929 and 0.864, respectively) (Extended Data Fig. 8a). To investigate which genes drive the observed epigenomic differences between medulla-projecting L5 ET neurons and other L5 ET neurons, we identified 1,380 (293) CH-DMGs between MOp (SSp)–medulla L5 ET neurons and at least one of the other ET projections (Fig. 4d, e, Supplementary Table 6). Among these, 180 CH-DMGs were identified in both MOp–medulla and SSp–medulla neurons (examples highlighted in Fig. 4d, e), which suggests a general regulatory mechanism that may be shared by different cortical regions. Accordingly, models trained in either MOp or SSp to distinguish neurons projecting to the medulla usually performed well when tested in the other region (Extended Data Fig. 8b). Similar enrichment of medulla-projecting neurons in subpopulations of L5 ET neurons has been reported in ALM using single-cell RNA sequencing (scRNA-seq) on cells labelled by retrograde injections (retro-seq)[6]. To compare these observations, we used gene body mCH as a proxy for gene expression to integrate our L5 ET epi-retro-seq data with the ALM retro-seq data. Joint *t*-SNE analysis showed that the medulla-projecting

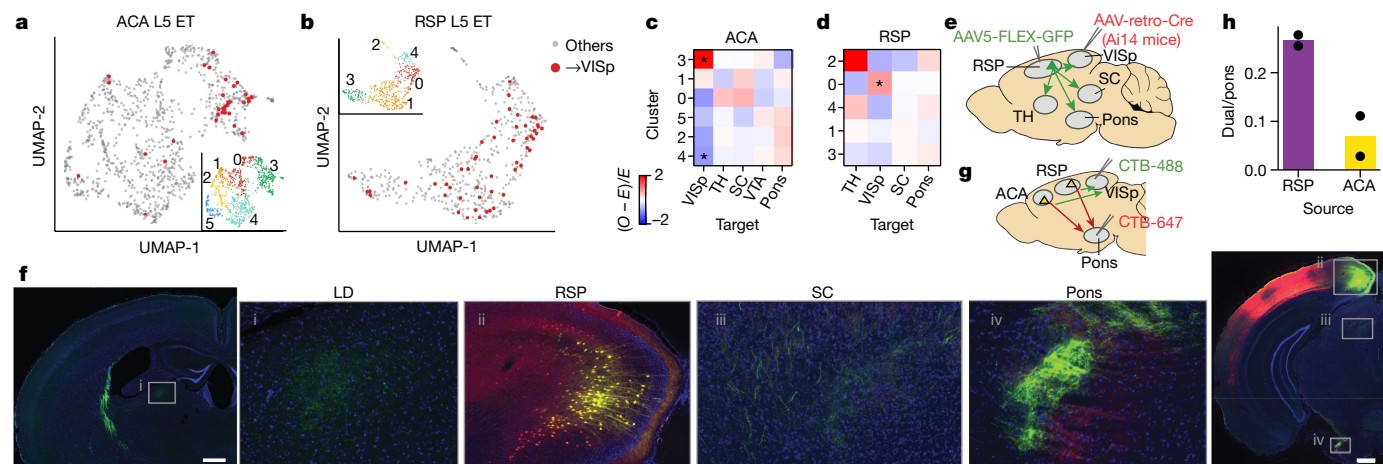

**Fig. 5 | A L5 ET neuron type that projects to both ET and cortical targets.**
**a**, **b**, UMAP embedding of ACA (**a**) or RSP (**b**) L5 ET neurons (*n* = 1,131 or 516)
using mCH in 100-kb bins, coloured by projection targets (ACA–VISp or RSP–
VISp in red, *n* = 36 or 51) or clusters (inset). **c**, ACA–VISp neurons were enriched
in ACA L5 ET cluster 3 and depleted from cluster 4. **d**, RSP–VISp neurons were
enriched in RSP L5 ET cluster 0. Asterisks in **c** and **d** indicate FDR < 0.05.
**e**, Illustration of the anatomical experiment to validate the existence of the L5
ET + CC cell type. **f**, VISp neurons at the AAV-retro-Cre injection site were
labelled by tdTomato (red). RSP–VISp neurons were labelled with GFP (green),
and RSP–VISp neurons at the AAV5-FLEX-GFP injection site were labelled
with both tdTomato and GFP (yellow; inset 'ii'). Scale bars, 500 µm (low
magnification). LD, laterodorsal thalamic nucleus. **g**, Illustration of injections
of dual retrograde tracers (CTB-488 and CTB-647) into the pons and VISp.
**h**, Proportion of double-labelled neurons (projecting to both pons and VISp)
among all neurons projecting to the pons in different sources. *n* = 2 biological
replicates are shown as individual points.

L5 ET neurons were enriched in the same cluster (Extended Data Fig. 9).
The *Slco2a1* marker gene of the ALM medulla-projecting cluster is
hypomethylated in MOp–medulla but not in SSp–medulla neurons
(Extended Data Fig. 9h). We identified *Astn2* as a marker gene for the
medulla-projecting L5 ET cluster in both the MOp and SSp (Extended
Data Fig. 9i). ASTN2 mediates the recycling of neuronal cell adhesion
molecule ASTN1 in migrating neurons[18], and its deletion has been
associated with neurodevelopmental disorders[19]. This suggests that,
compared with other L5 ET neurons, neurons projecting to the medulla
have distinct molecular properties, and these distinctions are probably
shared across several cortical regions.

In addition to the medulla-projecting L5 ET neurons, we also observed
differences in genome-wide mCH profiles between other ET projec-
tions. For example, L5 ET neurons in AI were segregated into five clus-
ters (Fig. 4f), and AI–pons and AI–superior colliculus neurons were
enriched in different clusters (Fig. 4g, h, Extended Data Figs. 7c, 8c).
By contrast, AI–pons and AI–thalamus neurons were enriched in simi-
lar clusters (Extended Data Figs. 7c, 8c). Analysis of gene body mCH
identified 145 CH-DMGs that were differentially methylated between
AI–superior colliculus neurons versus AI–pons, whereas most of them
had similar methylation patterns between AI–pons and AI–thalamus
neurons (Fig. 4i, Supplementary Table 6). Together, the results suggest
that AI–pons neurons are more distinct from AI–superior colliculus
neurons and are similar to AI–thalamus neurons.

In contrast to the conservation across cortical areas ALM, MOp
and SSp for differences related to projections to medulla, differ-
ences between pons-projecting and superior colliculus-projecting
neurons were not conserved across all cortical areas. The prediction
model trained to distinguish between pons-projecting versus superior
colliculus-projecting neurons performed well in distinguishing them
from cortical regions AI (AUROC = 0.939) and VIS (AUROC = 0.868),
but performed poorly in PTLp neurons (AUROC = 0.726) (Extended
Data Fig. 8a). The AUROC scores were correlated with the counts of
CH-DMGs identified between superior colliculus-projecting versus
pons-projecting neurons in the corresponding source regions (Spear-
man *r* = 0.683). We further hypothesized that in a cortical area where
more neurons project to both the pons and superior colliculus, the
epigenetic profiles of pons-projecting and superior colliculus-projecting

neurons are less distinguishable, and vice versa. To test this hypothesis,
we performed double retrograde labelling of the pons and superior
colliculus, and in each cortical source region we counted the number
of neurons labelled by only the tracer injected into the pons, only the
superior colliculus, or both (Supplementary Table 7). The highest per-
centage of double-labelled neurons was in the PTLp, and in general the
AUROC score from our model was negatively correlated with the pro-
portion of double-labelled cells across the cortical regions (Spearman
*r* = −0.829, *P* = 0.04) (Extended Data Fig. 8d). These correspondences
are weak, however, for most source regions, so the correlation is driven
primarily by the data from the PTLp.

## L5 ET + CC neurons

We noticed more than 30 neurons projecting to the VISp in L5 ET clus-
ters from the ACA and RSP datasets (Fig. 5a, b). Because neurons in
the L5 ET cluster are expected to project to ET targets, this finding
suggested that some L5 neurons might project to both cortical and
ET targets. These neurons were enriched specifically in one cluster in
ACA and RSP, respectively (FDR = 4.88 × 10^{-5} and 3.34 × 10^{-3}, Wald test)
(Fig. 5a–d). This type of cluster in both the RSP and ACA was marked by
the hypo-methylation of *Ubn2* (Extended Data Fig. 10a, top), a highly
expressed gene in visual systems, and many other genes also distin-
guished this cluster in either source (Extended Data Fig. 10a, bottom).

Although ET cells are generally thought to lack projections to other
cortical areas, there is some evidence for such cells from previous
studies[20–24]. To validate our findings anatomically for RSP–VISp ET neu-
rons in mice, we injected AAV-retro-Cre in the VISp and AAV-FLEX-GFP
(Cre-dependent GFP) in the RSP (Fig. 5e) or ACA (Extended Data
Fig. 10b) of three mice. This resulted in labelling of the complete
axonal and dendritic arbors of RSP–VISp or ACA–VISp neurons such
that their long-distance projections to locations other than VISp could
be assessed. For the RSP cases, we observed strong GFP labelling of axon
terminals in subcortical ET regions, including the thalamus, superior
colliculus and pons, in all three mice (Fig. 5f). For the ACA cases, axon
labelling in subcortical ET regions was weaker but still readily apparent
in the thalamus (Extended Data Fig. 10b). These results indicate that
single neurons in L5 of RSP and ACA can project simultaneously to

both cortical and subcortical ET targets in mice. Because these cells genetically cluster with L5 ET cells, we consider them a subtype of L5 ET cells that we refer to as 'L5 ET + CC'. We do not use the term 'L5 ET + IT' because many L5 ET neurons are known to project to another part of the telencephalon, the striatum.

To further assess and quantify the prevalence of L5 ET + CC cells in the ACA, RSP and other cortical areas, we performed dual injections of retrograde tracers into the pons (cholera toxin subunit B (CTB) Alexa Fluor 647) and the VISp (CTB Alexa Fluor 488) of two mice (Fig. 5g). Injections were made into topographic locations in pons known to receive input from ACA and RSP. Accordingly, overlapping retrogradely labelled neurons were observed in both ACA and RSP, allowing assessment of the proportion of double-labelled neurons within the overlap regions. Overlapping labels were also observed and quantified in higher visual cortical areas lateral and medial to VISp. A markedly high proportion (26.6%) of RSP neurons projecting to pons were double labelled (Fig. 5h, Supplementary Table 8). Substantial but smaller proportions were observed in the ACA (7.0%) (Fig. 5h, Supplementary Table 8) and lateral and medial higher visual areas (13.1% and 14.6%, respectively) (Extended Data Fig. 10c, Supplementary Table 8).

## Discussion

In this Article, we have quantitatively analysed and compared the methylation of mouse cortical neurons projecting to different cortical and subcortical targets. We identified differences between both IT neurons projecting to different cortical areas and between L5 ET neurons projecting to different ET targets. Cortical IT neurons that projected to different cortical targets varied in the extent of their epigenetic differences. Differences between projection target pairs were typically larger than differences between cortical source areas for any given pair of projection targets. Most distinct among the L5 ET neurons were those projecting to the medulla. This difference has been previously described for neurons in cortical area ALM[2], and we find that this difference is conserved across the additional cortical areas that we analysed, including the MOp and SSp. By contrast, differences between L5 ET neurons projecting to superior colliculus versus pons were more distinct in some cortical areas (such as AI) than in others (such as PTLp).

We found that a subpopulation of cortico-cortical RSP–VISp and ACA–VISp neurons clustered with L5 ET cells, in contrast to the expectation that L5 ET and IT cortico-cortical cells are distinct populations. This suggested that some L5 ET cells might project to cortical targets and this hypothesis was validated anatomically. Our anatomical experiments showed that RSP–VISp cells do project to many ET targets, including the thalamus, superior colliculus and pons, and we refer to this cell type as L5 ET + CC. Although we found CC projection neurons that clustered with L5 ET cells for only two of the twenty-six CC projections that we sampled, there remain many other combinations that we did not test. For example, our double retrograde labelling studies identified L5 ET + CC neurons in visual cortical areas that are lateral and medial to VISp. Furthermore, previous studies have described L5 ET + CC cells in primary and secondary motor cortex[21,22]. It is therefore likely that future studies will reveal L5 ET + CC neurons in additional cortical areas projecting to various combinations of ET and cortical targets.

Finally, this large-scale effort linking methylation status directly to the projection targets of mouse cortical neurons allowed us to identify differences between projection cell types in transcription factors linked to differentially methylated regions. These observations provide insight into genetic mechanisms that might contribute to the differences in morphology and function of these cell types. As we have shown, this large dataset also provides the opportunity to predict regulatory elements that might be harnessed in future studies to target transgene expression to these cell types.

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

## Methods

No statistical methods were used to predetermine sample size. The experiments were not randomized. The investigators were not blinded to allocation during experiments and outcome assessment.

### Experimental animals

All experimental procedures using live animals were approved by the Salk Institute Animal Care and Use Committee. The knock-in mouse line, R26R-CAG-loxp-stop-loxp-Sun1-sfGFP-Myc (INTACT) was used for most experiments[10] and they were maintained on a C57BL/6J background. Adult (42–49 day old) male and female INTACT mice were used for the retrograde labelling experiment. Adult C57BL/6J 'wild-type' mice were used for double-retrograde labelling experiments.

### Surgical procedures for viral vector and tracer injections

To label neurons projecting to regions of interest, injections of rAAV2-retro-Cre (produced by Salk Vector Core or Vigene, $2 \times 10^{12}$ to $1 \times 10^{13}$ viral genomes per ml, produced with capsid from Addgene plasmid 81070 packaging pAAV-EF1a-Cre from Addgene plasmid 55636) were made into both hemispheres of the INTACT mice. Mice were anaesthetized with either ketamine–xylazine or isoflurane, placed in a stereotaxic frame, and 0.1–0.5 µl of AAV was injected by pressure into stereotaxic coordinates corresponding to the desired projection target. A list of injection coordinates and volumes is provided in Supplementary Table 1. At least two male and two female mice were injected for each projection target. To label RSP or ACA neurons that project to VISp, VISp was injected with rAAV2-retro-Cre, and either RSP or ACA was injected with AAV-FLEX-GFP (Salk Vector Core) into 6 adult (3 RSP and 3 ACA) Ai14 mice. Therefore, RSP–VISp or ACA–VISp neurons, including their axonal projections, were selectively labelled with GFP. If RSP–VISp or ACA–VISp neurons also project to ET targets (L5 ET + CC neurons exist), GFP-labelled axons would be expected in subcortical ET targets such as the superior colliculus, pons and the thalamus.

### Assessment of double-retrograde labelling

To assess the double labelling of cortical cells projecting to the pons and/or superior colliculus, or projecting to pons and/or VISp, stereotaxic pressure injections of 0.1–0.2 µl of 0.25–0.5% of CTB Alexa Fluor 488 or 647 conjugated (Molecular Probes) were successfully made into the pons and the superior colliculus of 4 mice, or into the pons and VISp of 2 mice. Then, 6–7 days later, mice were perfused with PBS followed by 4% paraformaldehyde in PBS. Brains were removed and sectioned coronally at 40-µm thickness with a freezing microtome. Sections were mounted and imaged with a 20× epifluorescence objective and images assessed to identify single- and double-labelled neurons that were assigned to cortical areas. Sections with less than five labelled cells from either one of the injections were excluded, as were sections in which there were not at least ten labelled cells from one of the injections. Therefore, some cortical areas in which there was minimal or no overlap were not included. For each mouse, double-labelled cells were quantified for each region and expressed either as the proportion of double-labelled cell divided by the sum of all labelled cells (pons and superior colliculus), or as the proportion of double-labelled cells divided by the number of cells labelled from the pons (pons and VISp). Mean values from the four mice with CTB injections into the superior colliculus and pons are plotted in Extended Data Fig. 8d. Values from the two mice with CTB injections into the pons and VISp are shown in Fig. 5h and Extended Data Fig. 10c.

### Brain dissection

Approximately two weeks after the AAV-retro injection, brains were extracted from the 56–63-day-old INTACT mice, immediately submerged in ice-cold slicing buffer (2.5 mM KCl, 0.5 mM CaCl$_2$, 7 mM MgCl$_2$, 1.25 mM NaH$_2$PO$_4$, 110 mM sucrose, 10 mM glucose and 25 mM NaHCO$_3$) that was bubbled with carbogen, and sliced into 0.6-mm coronal sections starting from the frontal pole. From each mouse brain injected with AAV-retro, the slices were kept in the ice-cold dissection buffer, and selected brain regions (Supplementary Table 1) were manually dissected under a fluorescent dissecting microscope (Olympus SZX16), following the Allen Mouse Common Coordinate Framework (CCF), Reference Atlas, Version 3 (2015) (Extended Data Fig. 1). Olympus cellSens dimension 1.8 was used for image acquisition. The dissected brain tissues were transferred to pre-labelled microcentrifuge tubes, immediately frozen in dry ice, and subsequently stored at −80 °C.

### Nuclei preparation and single-nucleus isolation

For each dissected brain region, samples from two males and two females (except AI–pons, which were two male mice only) were pooled separately as biological replicates for nuclei preparation. The 2-ml glass tissue dounce homogenizer and pestles (Sigma-Aldrich D8938-1SET) were pre-chilled on ice. Nuclei were prepared using a modified protocol as previously reported[26]. In summary, the frozen brain tissues were transferred to the dounce homogenizer with 1 ml ice-cold NIM buffer (0.25 M sucrose, 25 mM KCl, 5 mM MgCl$_2$, 10 mM Tris-HCl (pH 7.4), 1 mM DTT (Sigma 646563), 10 µl of protease inhibitor (Sigma P8340)), with 0.1% Triton X-100 and 5 µM Hoechst 33342 (Invitrogen H3570), and gently homogenized on ice with the pestle 10–15 times. The homogenate was transferred to pre-chilled microcentrifuge tubes and centrifuged at 1,000$g$ for 8 min at 4 °C to pellet the nuclei. The pellet was resuspended in 1 ml ice-cold NIM buffer, and again centrifuged at 1,000$g$ for 8 min at 4 °C. The pellet was then resuspended in 450 µl of ice-cold NSB buffer (0.25 M sucrose, 5 mM MgCl$_2$, 10 mM Tris-HCl (pH 7.4), 1 mM DTT, 9 µl of protease inhibitor), and filtered through 40-µm cell strainer. The filtered nuclei suspension was incubated on ice for at least 30 min with 50 µl of nuclease-free bovine serum albumin (BSA) for at least 10 min, then incubated with GFP antibody, Alexa Fluor 488 (Invitrogen, A-21311, 1:500 dilution) and an anti-NeuN antibody (EMD Millipore MAB377, 1:300 dilution) conjugated with Alexa Fluor 647 (Invitrogen A20173). GFP$^+$NeuN$^+$ single nuclei were isolated using FANS on a BD Influx sorter with a 100-µm nozzle, and sorted into 384-well plates preloaded with 2 µl of digestion buffer for snmC-seq2[8] (20 ml digestion buffer consists of 10 ml M-digestion buffer (2×, Zymo D5021-9), 1 ml proteinase K (20 mg, Zymo D3001-2-20), 9 ml water, and 10 µl unmethylated lambda DNA (100 pg µl$^{-1}$, Promega, D1521)). The collected plates were incubated at 50 °C for 20 min then stored at −20 °C. BD Influx Software v.1.2.0.142 was used to select cell populations.

### snmC-seq2 library preparation

Nuclei from the same projection were combined in one 384-well plate for the library preparation. We assayed approximately 384 nuclei from each projection (except the MOp–SSp projection from which 768 nuclei were assayed). The bisulfite conversion and library preparation were performed following the detailed snmC-seq2 protocol as previously described[8]. The snmC-seq2 libraries were sequenced on Illumina Novaseq 6000 using the S4 flow cell 2 × 150 bp mode. Freedom EVOware v2.7 was used for library preparation, and Illumina MiSeq control software v.3.1.0.13 and NovaSeq 6000 control software v.1.6.0/Real-Time Analysis (RTA) v.3.4.4 were used for sequencing.

### Reads processing and quality controls

We used the cemba-data pipeline to generate allc files from fastq files (cemba-data.rtfd.io), as previously described[12]. Specifically, the fastq files were first demultiplexed into single cells and trimmed of Illumina adaptors and 10 bp on both sides with Cutadapt[27]. The reads were mapped to mm10 INTACT mouse genome using Bismark[28] with Bowtie2 aligner for each single end separately. The reads with MAPQ smaller than ten were excluded. Potential PCR duplicates were removed with Picard MarkDuplicates. The reads from two ends were then merged to generate allc files using call_methylated_sites function in methylpy[29].

The global mCCC level was used to estimate the non-conversion rate of bisulfite treatment. The cells with less than 500,000 non-clonal reads or non-conversion rate greater than 1% were removed from further analysis.

## Methylation data processing

For each single cell, we computed the methylated CH (mc) and total CH (tc) base calls of all 100-kb bins across the genome and all gene bodies annotated in GENCODE v.M10[30]. The autosomal bins that were covered by more than 100 base calls in greater than 95% of cells were used for further analysis. The autosomal genes that were covered by more than 100 base calls in greater than 80% of cells were used for further analysis.

## Computing posterior methylation levels

For each cell, we calculated the mean ($m$) and variance ($v$) of the mCH level across the 100-kb bins or genes. Then a beta distribution was fit for each cell $i$, in which the parameters were then estimated by:

$$\alpha_i = m_i\left(\frac{m_i(1-m_i)}{v_i} - 1\right)$$

$$\beta_i = (1-m_i)\left(\frac{m_i(1-m_i)}{v_i} - 1\right)$$

We then calculated the posterior mCH of each bin by:

$$\text{ratio}_{ij} = \frac{\alpha_i + \text{mc}_{ij}}{\alpha_i + \beta_i + \text{tc}_{ij}}$$

We normalized this rate by the global mean methylation of the cell by:

$$\text{global}_i = \frac{\alpha_i}{\alpha_i + \beta_i}$$

$$M_{ij} = \frac{\text{ratio}_{ij}}{\text{global}_i}$$

The values greater than 10 in $M$ were set to 10. After normalization, $M_{ij}$ is close to 1 when $\text{tc}_{ij}$ is close to 0.

## Identification of highly variable bins

Highly variable methylation features were selected on the basis of a modified version of the highly_variable_genes function from the SCANPY package[31]. In brief, because both the mean methylation level and the mean coverage of a feature (100-kb bin or gene) can affect dispersion of the methylation level[12], we grouped features that fall into a combined bin of mean and coverage, and then normalized the dispersion within each group. After dispersion normalization, we selected the top 2,000 features based on normalized dispersion for dimension reduction.

## Removing potential doublets

By plotting all cells on $t$-SNE, we noticed a cell population that was located in the centre of the plot and has a greater number of non-clonal reads than the others. To remove these potential doublets, we modified scrublet[32] to adopt it to methylation data. Specifically, we first simulated the doublet cells by randomly selecting two cells in our dataset and summed the methylation and total base calls of the two cells. Then the methylation levels of the simulated cells were computed using the posterior computing method. We simulated twice the number of doublets as the number of real cells. The top 2,000 highly variable features were selected for dimension reduction with principal component analysis (PCA) and the top 50 principal components were used to train a $k$-nearest neighbour (KNN) classifier ($k = 50$) to predict a doublet score

for each cell. On the basis of the histogram of doublet scores of real and simulated doublet cells, the cells with doublet score higher than 0.1 were removed from further analysis. After removing the potential doublets, 13,414 cells were kept for further analysis.

## Cell clustering and annotation

After removing potential doublets, the top 2,000 highly variable features were selected for dimension reduction with PCA. The top 50 principal components were used for $t$-SNE visualization and construction of KNN graph ($G$) with Euclidean distance ($k = 25$). We use $A$ to represent the connectivity of $G$, in which $A_{ij}$ is 1 if node $j$ is among the 25 nearest neighbours of node $i$, otherwise 0. The edge weights of $G$ were assigned as the jaccard distance of the connectivity matrix $A$. We ran Louvain clustering (https://github.com/taynaud/python-louvain) with resolution 1.2 to partition the cells into 31 clusters and merged these clusters into major cell subclasses based on known marker genes. Specifically, $Cux2^+ Rorb^-$ (hypomethylation in $Cux2$ gene body and hypermethylation in $Rorb$ gene body) was annotated as L2/3; $Cux2^+ Rorb^+$ was annotated as L4; $Cux2^- Rorb^+$ and $Deptor^+$ were annotated as L5 IT; $Sulf1^+$ and $Sulf2^+ Deptor^-$ were annotated as L6 IT; $Vat1l^+$ was annotated as L5 ET; $Foxp2^+$ was annotated as L6 corticothalamic; $Tle4^+ Foxp2^-$ was annotated as L6b; $Tshz2^+$ was annotated as near-projecting; $B3gat2^+$ was annotated as claustrum; $Slc6a1^+$ was annotated as inhibitory. The clusters with low global mCH level were annotated as non-neural cells, which were further confirmed by hypermethylation of $Mef2c$. The 11,827 cells within neuronal cell clusters were selected for further analysis.

## Inclusion criteria for confident target assignment

We implemented criteria to identify experiments in which artefacts could lead to inclusion of neurons that did not actually project to the intended AAV-retro injection site. Neurons failing these criteria were excluded from analyses requiring identification of projection targets but were included for analyses related to neuron sources. Close inspection of the distribution of cells sampled from each projection across subclasses revealed two types of artefact: (1) for some weak projections very few neurons were retrogradely labelled, resulting in small proportions passing FANS gating criteria and subsequent inclusion of high proportions of cells accepted from the edges of FANS gates ('gating artefact'); (2) AAV-retro injection pipettes targeting deep structures (for example, thalamus) passed through overlying cortical areas and directly labelled neurons rather than being taken up retrogradely from the intended target. This second artefact is apparent in previously published retro-seq data in which VISp IT neurons are prominent in putative cortico-tectal and cortico-pontine projection neuron populations (figure 3 and extended data figure 10 in Tasic et al.[6]). This suggests that injections passed through VISp, which directly overlies pons and tectum. In our experiments, injections to the superior colliculus and pons took oblique trajectories to minimize involvement of overlying cortical areas, but this was not possible for injections to the ventral tegmental area or thalamus.

Because FANS errors would be manifested in separate sorting runs, we assessed each FANS sorting case separately. To identify cases with high proportions of contaminating neurons (probably projecting to a different target than intended), for each FANS run, we counted the numbers of neurons that were observed in known on-target subclasses ($O_{\text{on}}$) and off-target subclasses ($O_{\text{off}}$). Assuming that the proportions of contaminated cells in each subclass would be similar to a sample without projection-type enrichment, we compared the observed counts to the counts from unbiased cortical samples[33] ($E_{\text{on}}$ and $E_{\text{off}}$) collected from the slices in Extended Data Fig. 1. The fold-enrichment was computed as $\frac{O_{\text{on}}E_{\text{off}}}{O_{\text{off}}E_{\text{on}}}$. A one-sided exact binomial test of goodness-of-fit was used to determine whether the enrichment of on-target cells was significant. Specifically, the $P$ value was computed as: $\Pr(X \geq O_{\text{on}}; n, p)$, in which

$$X \sim \text{Binomial}(n, p)$$

$$n = O_{\text{on}} + O_{\text{off}}$$

$$p = \frac{E_{\text{on}}}{E_{\text{on}} + E_{\text{off}}}$$

Neurons from cases in which the fold-enrichments were smaller than a threshold (see below) or the tests were not significant were categorized as having unknown projection targets. The expected values are different for ET targets than for IT (including striatum) targets, so the thresholds depend on the targets.

For each ET target, we considered L5 ET as on-target subclass and IT and inhibitory neurons as off-target. The thresholds for fold enrichment and FDR (Benjamini–Hochberg procedure) were 5 and 0.01, respectively. This eliminated 7 out of 101 ET target sorts (285 out of 5,364 cells). For IT targets, we considered IT as on-target subclasses and L6 corticothalamic plus inhibitory neurons as off-target. The thresholds for fold-enrichment and FDR (Benjamini–Hochberg procedure) were 3 and 0.05, respectively. This eliminated 30 out of 115 sorting cases (1,146 out of 6,463 neurons).

Note that these exclusion criterions are based on a simplified expectation of on target cell types, and the accuracy might be variable depending on the targets. For instance, when considering the neurons projecting to the striatum, considering L6 corticothalamic as off-target might overestimate the off-target cells and make the exclusion more stringent. In addition, because the filter was applied at FANS run-level, there could also be a small percentage of off-target cells from the included runs. This should be noticed when using these datasets. We included the cell type proportion of all projections in Extended Data Fig. 3c to help evaluate this potential noise.

### Neighbour enrichment score

The score was used to quantify the enrichment of cells that belong to the same category among the neighbours of each cell. A higher score means that the cells are more likely to form clusters with the cells belonging to the same category rather than in the other categories. The advantage of this score is that it only considers the local effect so that would remain high if the cells in a category form several different clusters that dissimilar with each other. The score was computed as follows. Euclidean distances between each pair of cells were computed using the first 50 principal components. For each cell, we found its 25 nearest neighbours in the same category, and $25r$ nearest neighbours from other categories, in which $r$ is the ratio between total number of cells in other categories and total number of cells in the same category. The AUROC scores using distances between the cell and these neighbour cells for distinguishing the categories were defined as the neighbour enrichment score of this cell. The methylation pattern of male and female mice are highly similar on autosomes; therefore, the two genders were treated as replicates in the analyses. When computing the score for targets, neurons with targets that were not confidently assigned were excluded. When computing the score for replicates, the AI–pons projection that only has one replicate was excluded.

### Pairwise prediction of the source and target regions

On the basis of the sources and targets, the neurons could be separated into groups. Each group contains the neurons projecting from a specific source to a specific target. To test the similarity of two groups of cells based on DNA methylation, we trained logistic regression models to predict the group label of each cell. The posterior of 100-kb bin or gene body mCH were used as features. We used two methods to split the cells into training and testing sets, one uses random selection of half of the cells for training and the other half for testing (computational replicates), the other is based on the gender of the mice the cells were

collected from (biological replicates). All results in the main figures were computed using the computational replicates, whereas the results using biological replicates are also provided in Extended Data Figs. 4 and 5. The results of corresponding comparisons were very similar between these two replicate-splitting methods. The AUROC score from cross-validation was used to measure the performance of the model. The higher AUROC value represents better ability of the model to predict the group label, which indicated the two groups had larger mCH differences and were more distinguishable. Sci-kit learn was used for model implementation.

When the groups being studied contained cells from different subclasses (for example, cortical-projecting neurons in one source), we upsampled the training set to ensure that it captures the group differences rather than the differences of cell distributions across subclasses. For example, when comparing neurons projecting to two different cortical targets, the subclass composition differences could make the model over-weight the features marking different subclasses. To get rid of this bias, we randomly repeated the neurons from the underrepresenting group and ensured the two groups had the sample number of training samples in each subclass. The models were then trained and tested in the same setting as mentioned above.

Several reasons could contribute to a low prediction performance. Biological reasons are as follows. (1) Some neurons make projections to several targets simultaneously. These could result in the neurons being captured by several retrograde labelling experiments of different targets. It would be impossible to predict a single label with our pairwise models for this type of neuron. (2) Some neurons could project to different target regions but have tiny epigenetic differences. To systematically distinguish between (1) and (2), other anatomic and genetic validation are still needed.

Technical reasons are as follows. (3) The epigenetic differences between neurons projecting to different targets varies across replicates. (4) The contamination levels of some projections are relatively high, which makes larger noise and hinders the ability of the models to capture real signals. (5) The sample sizes of some projections are small, which make the learning more challenging. (6) The models are not powerful enough to capture the complex differences between projections.

In this study, male and female mice were treated as biological replicates after removing sex chromosomes. Although methylation patterns of autosomes are similar, differences between genders or individuals might still exist. The small differences of performances between data-splitting methods (based on computation or biological replicates) might suggest a less notable effect contributed by (3) in those samples. If the cross-source or layer predictions (described below) performed better than the within source or layer models, we would suspect that shared differences between neurons projecting to different targets exist across sources or layers, and the major reason for lower accuracies of within source or layer models might be (4) or (5). Elimination of contaminated FANS runs decreases the potential influence by (4), although there are still contaminated cells included in the dataset. To evaluate the potential limitation of (6), more carefully curated models, and accordingly more samples, would be required. Thus, given all these factors, we are generally more confident in the distinguishable target pairs when training and testing sets were split based on both computational and biological replicates. The interpretation of comparisons without biological replicates and the indistinguishable pairs would need to be more careful and are not involved in the major conclusions in this manuscript. Our Article aims to provide a general view across several sources and targets. More detailed understanding of specific projections would require larger scale profiles on those specific projection types.

### Cross source prediction

The logistic regression models were trained to predict the projection targets in one source and tested in the other source. The training and testing sets came from either the biological or computational

replicates. When using biological replicates, the final AUROC scores were the average of AUROCs by training in male mice in one source and testing in female mice in another source, and by training in female mice in the first source and testing in male mice in the second source. For cortical targets, we upsampled the training set as stated above.

Note that when the models were training only in one source, they would not necessarily capture the shared features across sources to distinguish neurons projecting differently even if some shared differential features exist. However, when more differential features are shared across sources, the models are more likely to select the shared ones. Thus, the low performance in the analysis might indicate that there are less differential features shared across sources and the models majorly selected the differential features specific to one source but not another source, rather than representing none of the differential features are the same between the two sources. By contrast, the high performances usually indicate that more differential features are shared between sources. Similar interpretation applies to the cross layer prediction in the next section.

### Cross layer prediction
This analysis was specifically for CC projection neurons to study whether the mCH differences between projection neurons were shared or distinct across layers. The logistic regression models were trained to predict the projection targets in all but one layer and tested in the one layer left out during training. The training and testing sets were split based on either computational or biological replicates as stated above.

### Identification of CH-DMGs
Wilcoxon rank-sum tests and $t$-tests were widely used to identify differential genes in single-cell studies[31], which consider each cell as an independent sample. However, the cells from the same replicate, individual or batch would be more similar than the cells from different ones. Therefore, considering all cells as independent samples would overestimate the statistical power in single-cell data. To address this problem and take the replicate-level variation into consideration, we used a linear mixed model for the differential analysis and performed paired-wise comparisons between groups. The posterior mCH levels of 12,261 autosomal genes after coverage filters were used for these analyses. The posterior gene body mCH was used as dependent variables. Each individual mouse was considered as a random effect. The global mCH levels and the gender of the mice were considered as fixed effects. Other fixed effects were determined on the basis of the comparison. Specifically, for DMGs between L5 ET clusters:
Gene_mCH ~ cluster + gender + global_mCH + (1 | mouse)
For DMGs between cortical targets in each source:
Gene_mCH ~ target + cluster + gender + global_mCH + (1 | mouse)
For DMGs between ET targets in each source:
Gene_mCH ~ target + gender + global_mCH + (1 | mouse)
Each gene was tested separately, and a two-sided Wald test was performed to estimate the $P$ value for the effect being tested. FDR was computed for each pair of groups with the Benjamini–Hochberg process. The fold change of each gene was computed by the average mCH across cells in one group divided by the average mCH across cells in the other group, with pseudo-counts of 0.1. The criterions for significance when testing different variables were distinct and shown as follows. For DMGs between L5 ET clusters: absolute log-transformed fold change greater than log1.5 and FDR smaller than 0.01. For DMGs between IT targets or between ET targets in each source: absolute log-transformed fold change greater than log1.25 and FDR smaller than 0.01.

### GO enrichment analysis
GO enrichment analysis was performed using the web server at http://geneontology.org/. The 12,261 genes that passed the coverage threshold mentioned above were used as background, and binomial tests were used to select the significant biological processes related to each

DMG list. Note that GO names are nomenclature that summarize many complex relationships between genes and their function, so we do not expect that these analyses can be used to directly infer how a particular gene contributes to neuronal function in a specific context.

### Identification of CG-DMRs
To identify DMRs, we merged the allc files of individual cells assigned to the same cluster to create a pseudo-bulk allc table for each cluster. Then we selected all the CG sites and combined the methylation on two DNA strands for each CpG site. We run methylpy[29] DMRfind to identify the DMRs and require the DMRs to contain at least two differentially methylated CpG sites (DMS).

### Inference of crucial transcription factors with PageRank
The method was modified from Taiji[34] to integrate the information of both gene body and regulatory regions. The 537 motifs in JASPAR 2018 non-redundant core vertebrate database[35] were used for these analyses. We scanned each of the motifs against the mm10 INTACT mouse genome with fimo[36] and $P$-value cutoff as $1 \times 10^{-5}$. The DMRs between clusters were expanded 100 bp on both sides, and the ones overlapping with motifs were assigned to the corresponding transcription factor. The DMRs were also assigned to the potential genes they regulated using GREAT[37]. The transcription factors were then linked with the target genes based on these DMRs that links to both the upstream transcription factors and the downstream genes. A gene regulation network was constructed where the nodes represented the genes and edges represented the links between transcription factor genes and target genes.

To assign weights to the edges and initiate the node importance, the normalized $n_{cluster} \times n_{gene}$ methylation matrix ($M$) were min-max normalized across genes within each cluster to 0–1 by

$$N_{ij} = \frac{M_{ij} - \min_{0 < j' \le n_{gene}} M_{ij'}}{\max_{0 < j' \le n_{gene}} M_{ij'} - \min_{0 < j' \le n_{gene}} M_{ij'}}$$

and $1 - N_{ij}$ was used as the predicted expression of each gene in cluster $i$. The predicted expressions of all genes were used as starting importance $I_0$. Then we used a $n_{gene} \times n_{gene}$ matrix $A$ to represent the adjacency matrix of transcription factor–gene regulation network, in which $A_{ij}$ was assigned as the predicted expression level of gene $i$ if gene $i$ is a transcription factor. To ensure an undirected propagation, we used $B = A + A^T$ as the final adjacency matrix. $B$ was normalized by row into the transition matrix $P$ by

$$P_{ij} = \frac{B_{ij}}{\sum_{j'=1}^{n_{gene}} B_{ij'}}$$

Next we performed a diffusion step of the PageRank scores through the network. For iteration $t$, the PageRank scores were computed by

$$I_t = (1 - \text{rp})PI_{t-1} + \text{rp}I_0$$

in which rp represents a restart probability to balance the global and local effect of the propagation on the network. The diffusion step was stopped when $|I_t - I_t| < 10^{-5}$.

### Clustering of L5 ET cells in each source region
L5 ET neurons from epi-retro-seq and unbiased snmC-seq were combined in this analysis. After the same process as clustering all cells to derive posterior mCH level and select highly variable features, the first 30 principal components were used for computing KNN ($k = 15$) and Louvain clustering. The resolutions used for source regions were 1.6 for MOp, AI, AUD and RSP; 2.0 for SSp and PTLp; 1.0 for VISp; and 2.5 for ACA. The resolutions were determined on the basis of visually examining the cluster numbers and projection enrichment.

To confirm that there were epigenetic features distinguishing the clusters, we computed the differentially methylated 100-kb bins (DMBs) across all pairs of clusters using two-sided Wilcoxon rank-sum tests. The bins were defined as differential if the absolute log-transformed fold change between clusters was greater than log 1.5 and the FDR of the test smaller than 0.01. We also used AUROC > 0.85 and area under precision/recall curve (AUPR) > 0.6 to define DMBs, which provided similar results. Two clusters in RSP that had less than 5 DMBs were merged.

## Tests of projection enrichment in clusters

As described above, the cells from the same replicate would be more similar, and considering all cells as independent samples will overestimate the statistical power in single-cell data. Therefore, we used linear mixed models to test for significant enrichment of particular projections in each cluster, considering the mouse where the cells came from. The cluster was used as dependent variables. Each individual mouse was considered as a random effect. The projection target was considered as fixed effects.

[Cluster ~ target + (1 | mouse)]

Each projection target and each cluster were tested separately, and two-sided Wald tests were performed to estimate the $P$ value for the effect being tested. FDR was computed for each source with the Benjamini–Hochberg process. '(Observed – expected)/expected' in the enrichment matrices were computed using the same method as in Pearson's chi-square test.

## Integration of epi-retro-seq and retro-seq

Single-cell transcriptomic data from Tasic et al.[6] was downloaded from NCBI Gene Expression Omnibus (GEO) accession GSE115746. Then, 365 cells within clusters of 'L5 PT ALM *Npsr1*', 'L5 PT ALM *Slco2a1*' and 'L5 PT ALM *Hpgd*' were selected for integration analysis. The raw data was preprocessed using SCANPY[31]. Specifically, the read counts were normalized by the total read counts per cell and log transformed. The top 10,000 highly variable genes were identified and $z$-score scaled across all the cells. For methylation data, the posterior methylation levels of 12,261 genes in the 4,176 L5 ET cells were $z$-score scaled across all the cells and used for integration. We used Scanorama[38] to integrate the $z$-scored expression matrix and minus $z$-scored methylation matrix with sigma equal to 100.

## Overlap score

Overlap score quantifies the similarity of the distributions of two groups of cells across clusters, in which higher scores represent the two groups are more likely to be co-clustered. The scores were computed using the same method previously described[7]. Specifically, a $n_\mathrm{group} \times n_\mathrm{cluster}$ matrix $C$ was first computed, in which $C_{ik}$ represents the number of group $i$ cells in cluster $k$. $C$ was normalized by row to $D$, and the overlap score between group $i$ and group $j$ was defined as $\sum_{k=1}^{n_\mathrm{cluster}} \min(D_{ik}, D_{jk})$.

## Reporting summary

Further information on research design is available in the Nature Research Reporting Summary linked to this paper.

## Data availability

Single-cell raw and processed data included in this study were deposited to the NCBI Gene Expression (GEO) SRA with accession number GSE150170 and the NeMO ftp archive: http://data.nemoarchive.org/biccn/lab/callaway/projection/sncell/. Another dataset used in this study includes the JASPAR motif database (http://meme-suite.org/db/motifs) and retro-seq data from GSE115746.

## Code availability

The code for all of the analyses can be found at https://github.com/zhoujt1994/EpiRetroSeq2020.git.

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

**Acknowledgements** We thank K. Zhang for advice on the PageRank algorithm, and J. R. Dixon for comments. We are grateful to M. Nunn for help with management of the project. This work is supported by NIMH U19MH114831 to E.M.C. and J.R.E. and by NIMH R01MH063912 and NEI R01EY022577 to E.M.C. M.A.K. is supported by NEI F31 EY028853. The Flow Cytometry Core Facility of the Salk Institute is supported by funding from NIH-NCI CCSG: P30 014195. J.R.E. is an investigator of the Howard Hughes Medical Institute.

**Author contributions** Research design: E.M.C., Z.Z, M.M.B., J.R.E., J.Z., X.J. and K.-F.L. Data collection: Z.Z., Y.P., A.C.R., E.W., C.-T.L., M.A.K., A.D.F., P.A.M, A.B., A.I.A., M.V., L.B., C.F., J.R.N., R.G.C., M.R., M.W.J., T.I.-C., B.D., J.B.S., C.O., A.P.-D. and M.M.B. Data analysis: J.Z., Z.Z., P.T., E.M.C., M.A.K., A.D.F., H.L. and S.-Y.N. Data archive/infrastructure: E.A.M., Z.Z., Y.P., A.C.R. and A.B. Research coordination: Z.Z., E.M.C., J.R.E., M.M.B., Y.P., X.J., E.W., C.-T.L., E.A.M. and K.-F.L. Writing manuscript: J.Z., Z.Z., E.M.C., P.T., J.R.E., E.A.M. and M.M.B.

**Competing interests** J.R.E. serves on the scientific advisory board of Zymo Research Inc.

**Additional information**
**Correspondence and requests for materials** should be addressed to J.R.E. or E.M.C.

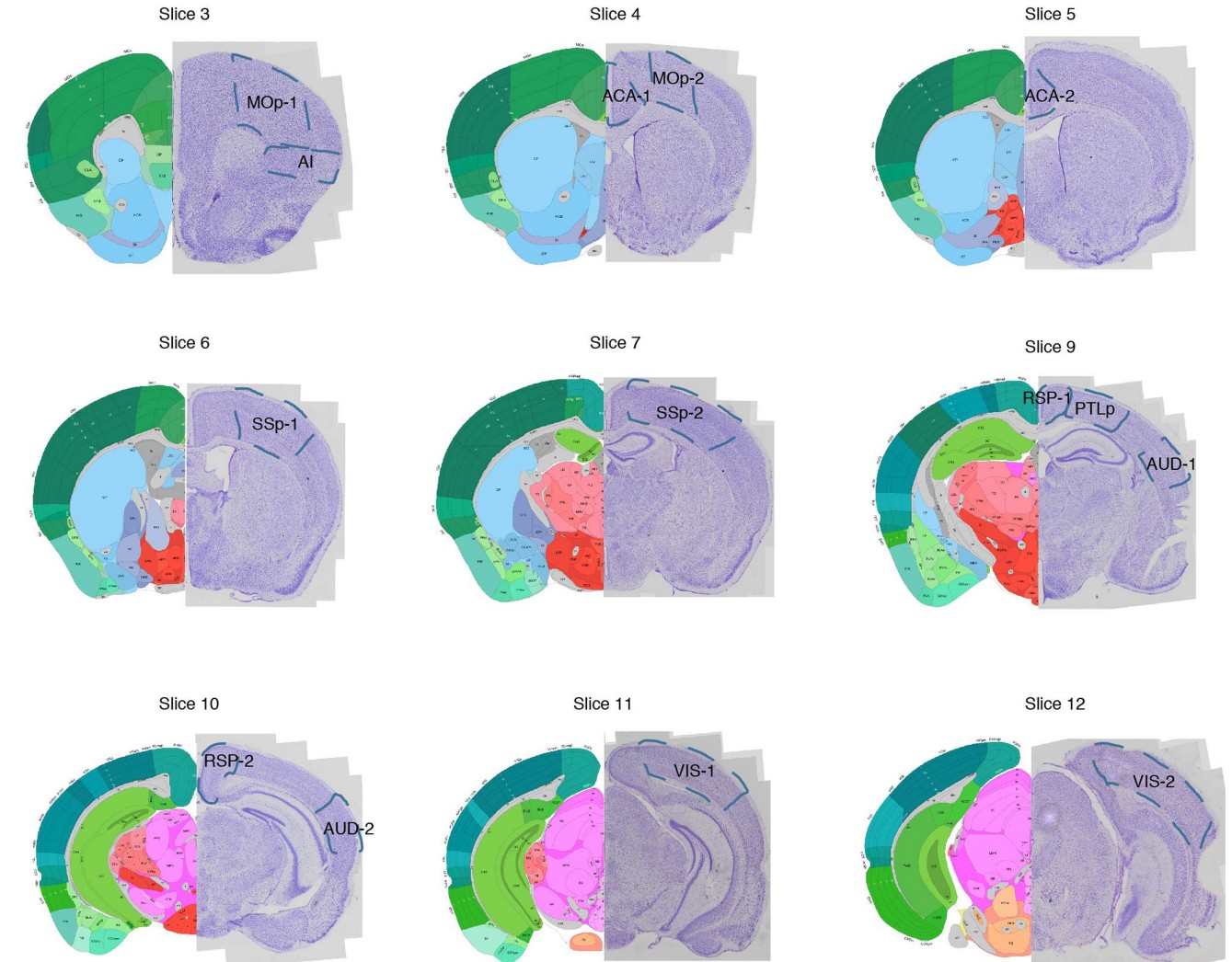

**Extended Data Fig. 1 | Source region dissection maps.** The posterior views of dissected slices are shown. The slices correspond to Allen Reference Atlas level 33–39 (slice 3), 39–45 (slice 4), 45–51 (slice 5), 51–57 (slice 6), 57–63 (slice 7), 69–75 (slice 9), 75–81 (slice 10), 81–87 (slice 11) and 87–93 (slice 12), respectively.

All brain atlas images were created based on Wang et al.[25] and © 2017 Allen Institute for Brain Science. Allen Brain Reference Atlas. Available from: http://www.atlas.brain-map.org.

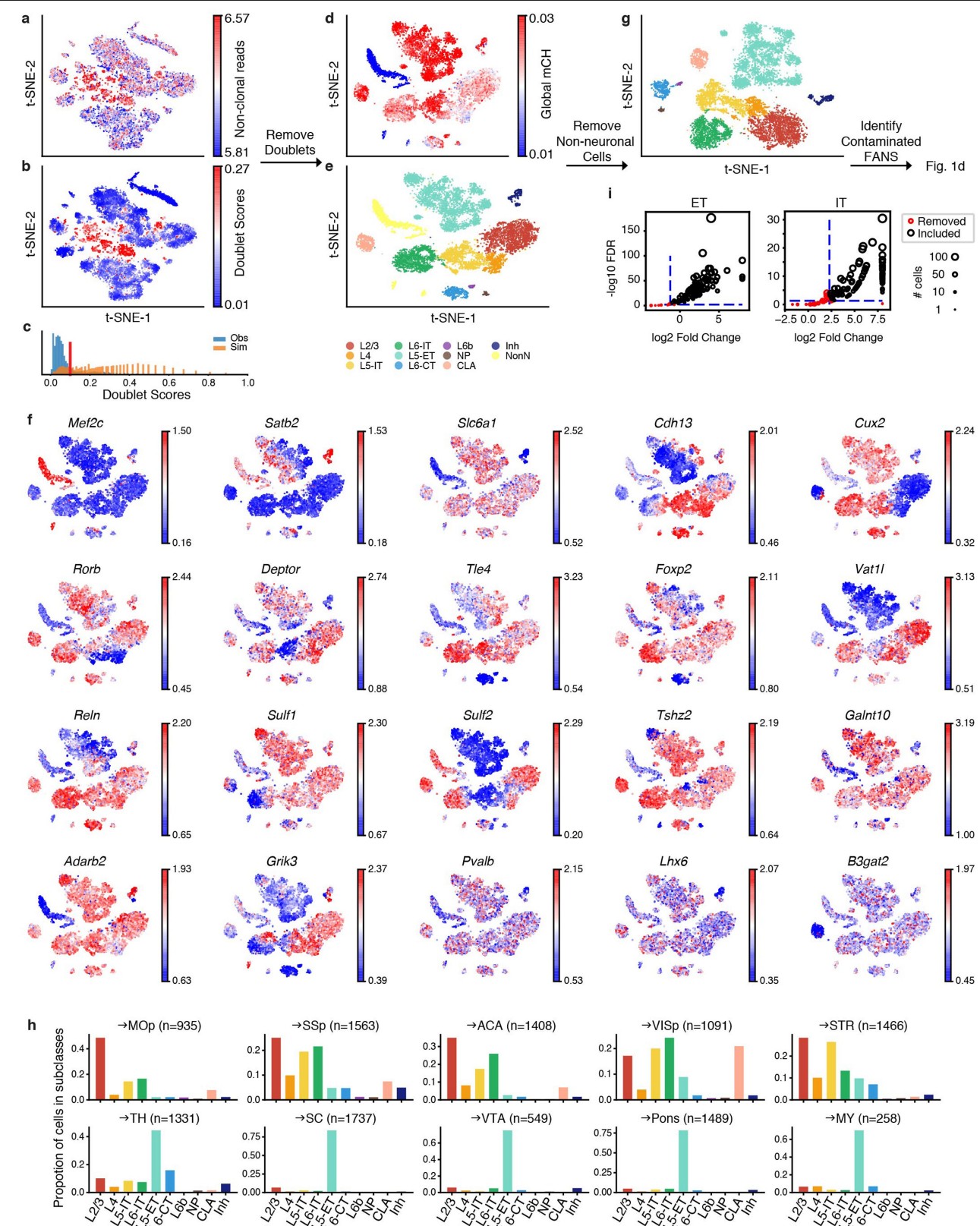

**Extended Data Fig. 2 | Removing potential doublets and non-neuronal cells. a**, **b**, *t*-SNE of cells after quality control (*n* = 16,971) coloured by number of non-clonal reads (**a**) and predicted doublet scores (**b**). **c**, Distribution of doublet scores for real cells (blue) and simulated doublets (orange). **d**–**f**, *t*-SNE of cells after removing doublets (*n* = 13,414), coloured by global mCH (**d**), subclass (**e**), or normalized gene-body mCH level of known cell type gene markers (**f**). Cells with low global mCH level are usually non-neuronal cells. **g**, *t*-SNE of single neurons (*n* = 11,827) coloured by subclass. **h**, Proportion of single neurons in each subclass for each projection. **i**, The scatter plots for filtering FANS runs with high contamination. Each dot represents a single run (*n* = 101 left, 115 right), and the size of the dot represents the number of on-target cells selected by the run.

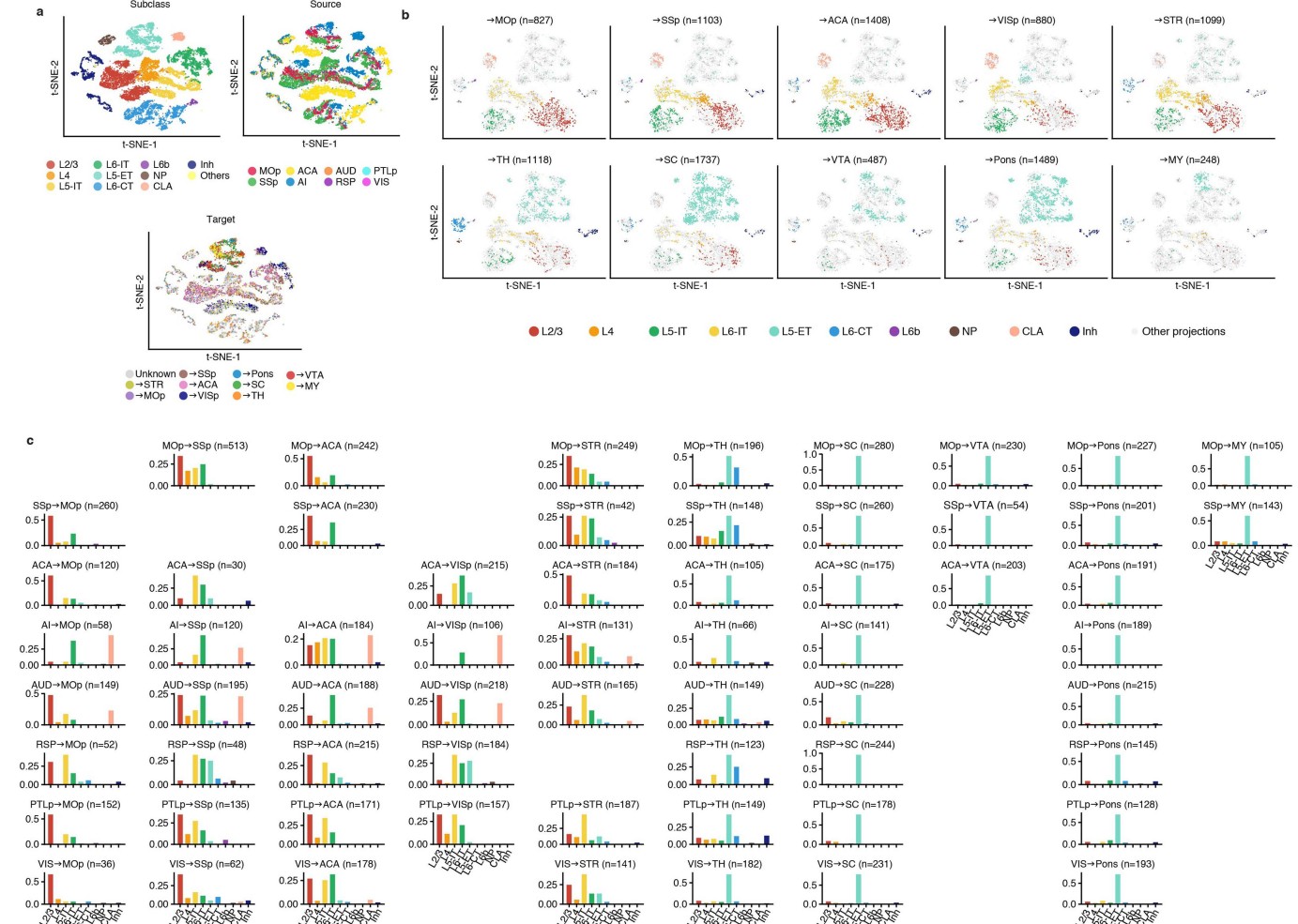

**Extended Data Fig. 3 | Cell type composition of all projections. a**, Joint *t*-SNE of neurons profiled by epi-retro-seq (*n* = 6,362) and unbiased snmC-seq2 (*n* = 15,782, without enrichment of projections) from MOp, SSp, ACA and AI, coloured by subclass (top left), source region (top right), and projection targets in epi-retro-seq (bottom). **b**, *t*-SNE of neurons (*n* = 11,827) projecting to each IT

target (top) and ET target (bottom). The cells projecting to the target were coloured by subclass and cells that project to all other targets or whose target was not confidently assigned were greyed. **c**, The proportion of cells projecting from each source (row) to each target (column) in all subclasses.

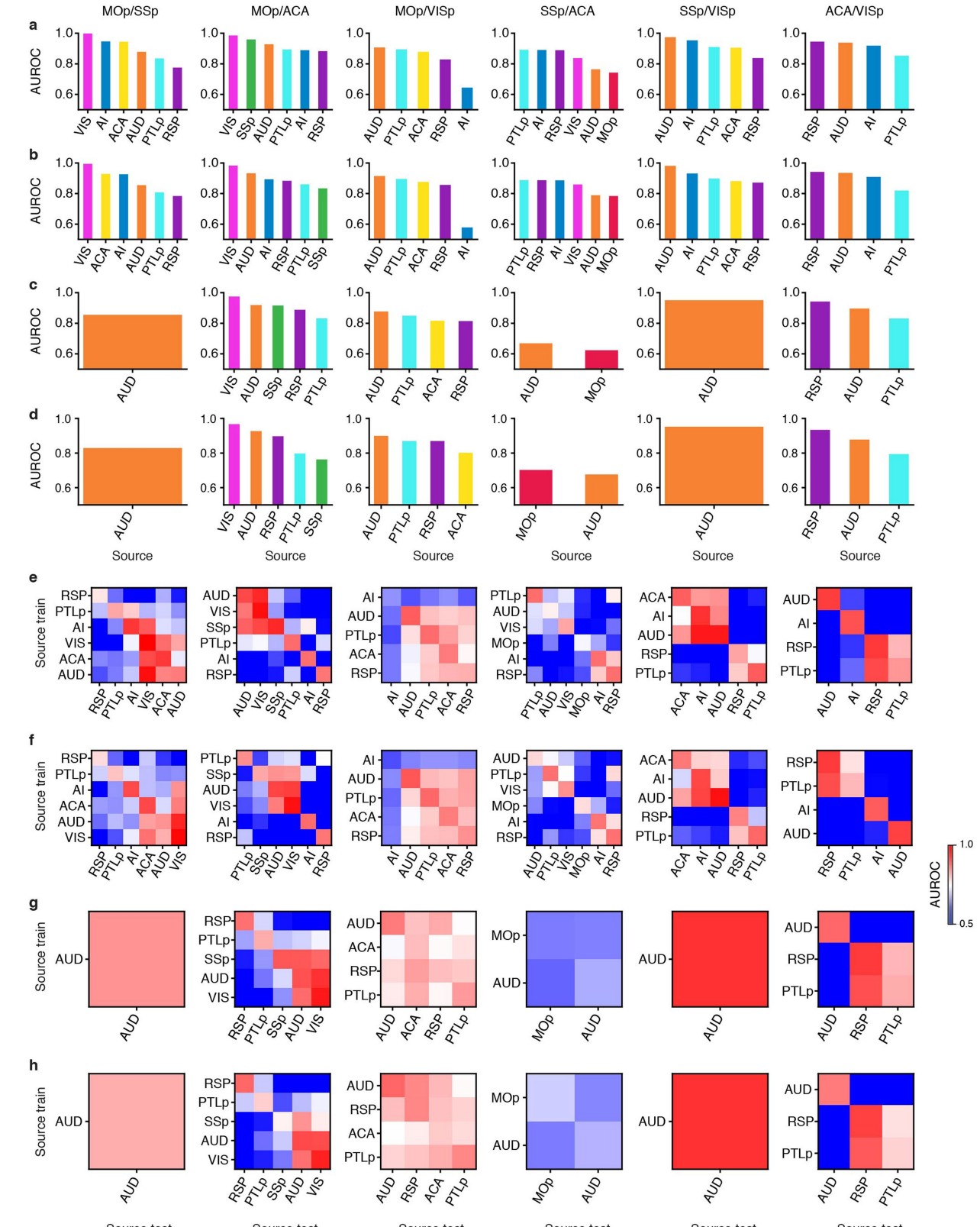

**Extended Data Fig. 4 | AUROC of cortical target pairs within and across sources. a–h,** AUROC scores of models trained and tested in the same source (**a–d**), or of models tested in all sources after being trained in each one of them (**e–h**). Gene body (**a, c, e, g**) or 100-kb bin (**b, d, f, h**) mCH was used as a feature. The training and testing sets were randomly split (**a, b, e, f**) or split based on biological replicates (**c, d, g, h**). The values in **a–d** correspond to the diagonals of **e–h** but ordered decreasingly.

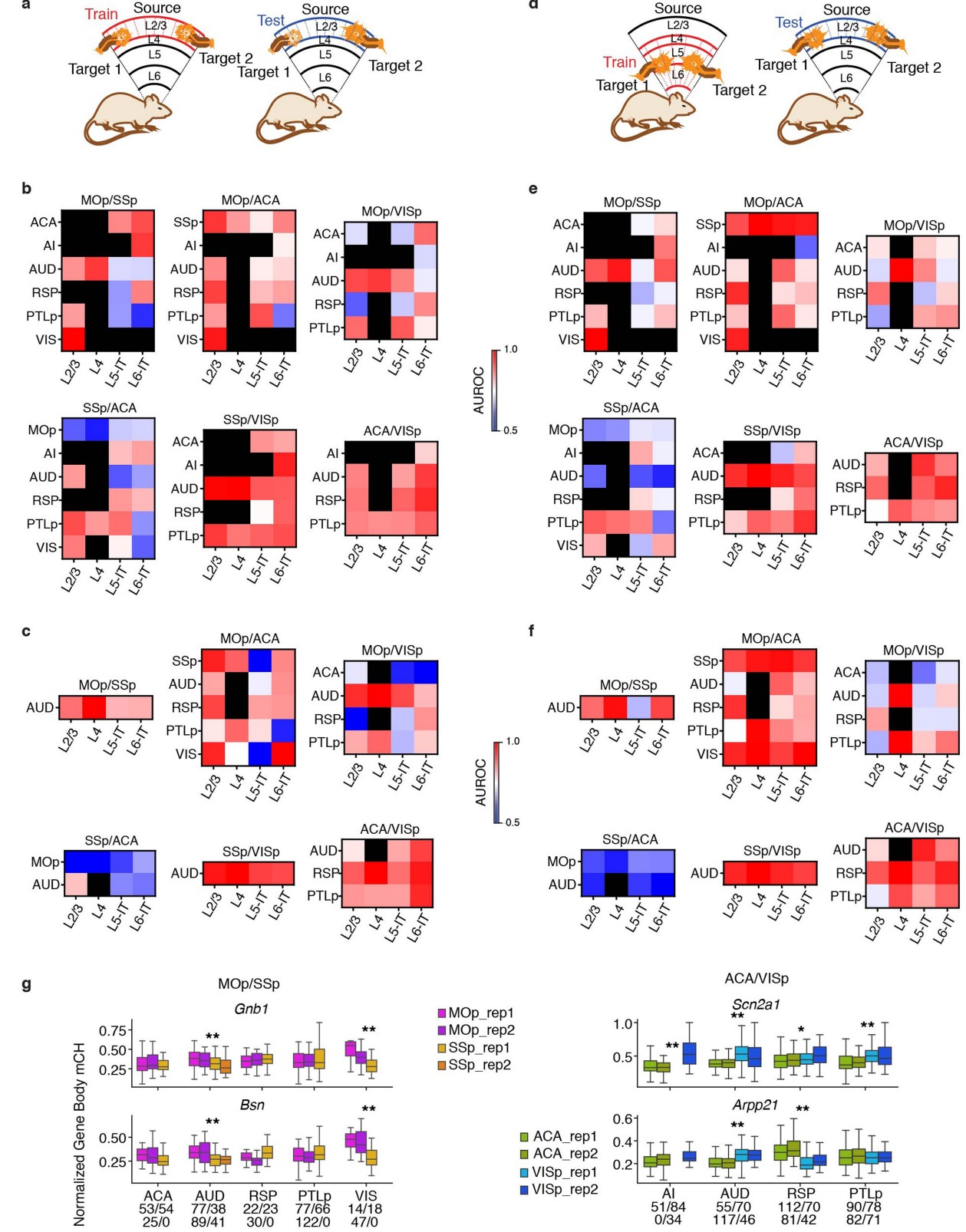

**Extended Data Fig. 5 | AUROC of cortical target pairs within and across layers. a, d,** Demonstration of training and testing data for within layer prediction (**a**) and cross layer prediction (**d**). In **a**, the models were trained and tested in the same layer with different cells. In **d**, the testing sets were as in **a**, but the models were trained in all other layers. **b, c, e, f,** AUROC of within-layer prediction (**b, c**) or cross-layer prediction (**e, f**). The training and testing sets

were randomly split (**b, e**) or split based on biological replicates (**c, f**). Gene level mCH were used for all the predictions. **g,** Box plots of example CH-DMGs between MOp versus SSp-projecting neurons (left), or between ACA versus VISp-projecting neurons (right). The sample sizes are shown below the *x* axis. *FDR < 0.1, **FDR < 0.01. Box plots are as in Fig. 1.

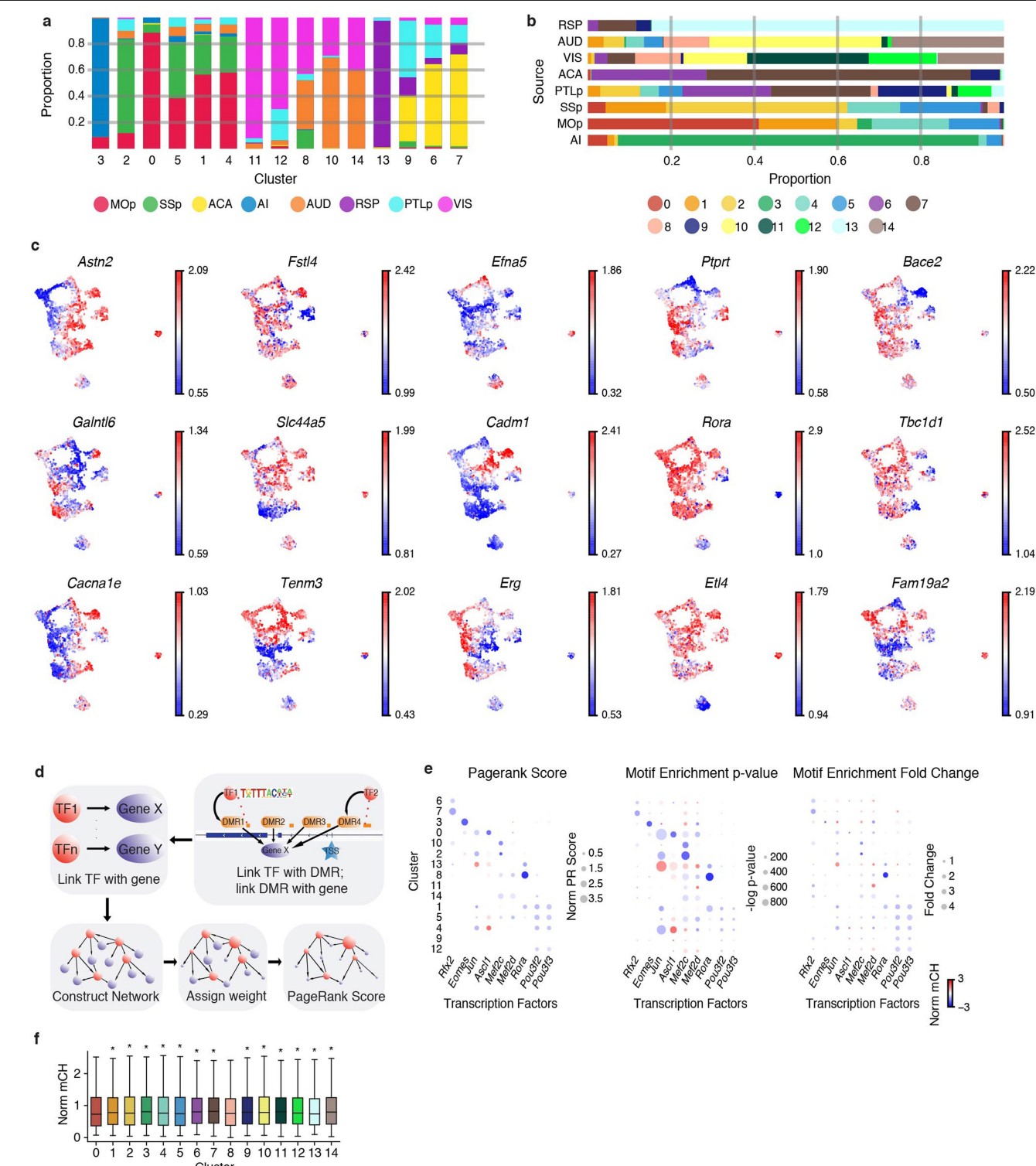

**Extended Data Fig. 6 | Signature genes and transcription factors of L5 ET clusters. a**, Proportion of cells from all sources in each cluster. **b**, Proportion of cells in all clusters from each source. **c**, *t*-SNE of L5 ET cells (*n* = 4,176) coloured by the normalized gene-body mCH level of cluster gene markers. **d**, Workflow of the PageRank algorithm to infer crucial transcription factors. **e**, Gene body mCH (colour) against PageRank score (size, left), motif enrichment *P* value (size, middle), and motif enrichment fold-change (size, right) for the example transcription factors in all L5 ET clusters. *P* values were computed by Homer

using one-sided binomial tests. **f**, Gene body mCH in all clusters of *Rora* target genes identified in cluster 8 (*n* = 3,299). Significances were determined by comparing cluster 8 with each of the other clusters (two-sided Wilcoxon signed-rank test, Benjamini–Hochberg FDR). *FDR < 1 × 10⁻². FDR for all boxes are: 0.60, $1.95 \times 10^{-25}$, $3.56 \times 10^{-12}$, $5.24 \times 10^{-29}$, $1.57 \times 10^{-10}$, $8.44 \times 10^{-9}$, $2.94 \times 10^{-32}$, $3.56 \times 10^{-41}$, 1.0, $1.16 \times 10^{-35}$, $5.85 \times 10^{-29}$, $2.28 \times 10^{-42}$, $1.47 \times 10^{-28}$, $6.42 \times 10^{-3}$ and $1.50 \times 10^{-26}$ (left to right). Box plots are as in Fig. 1.

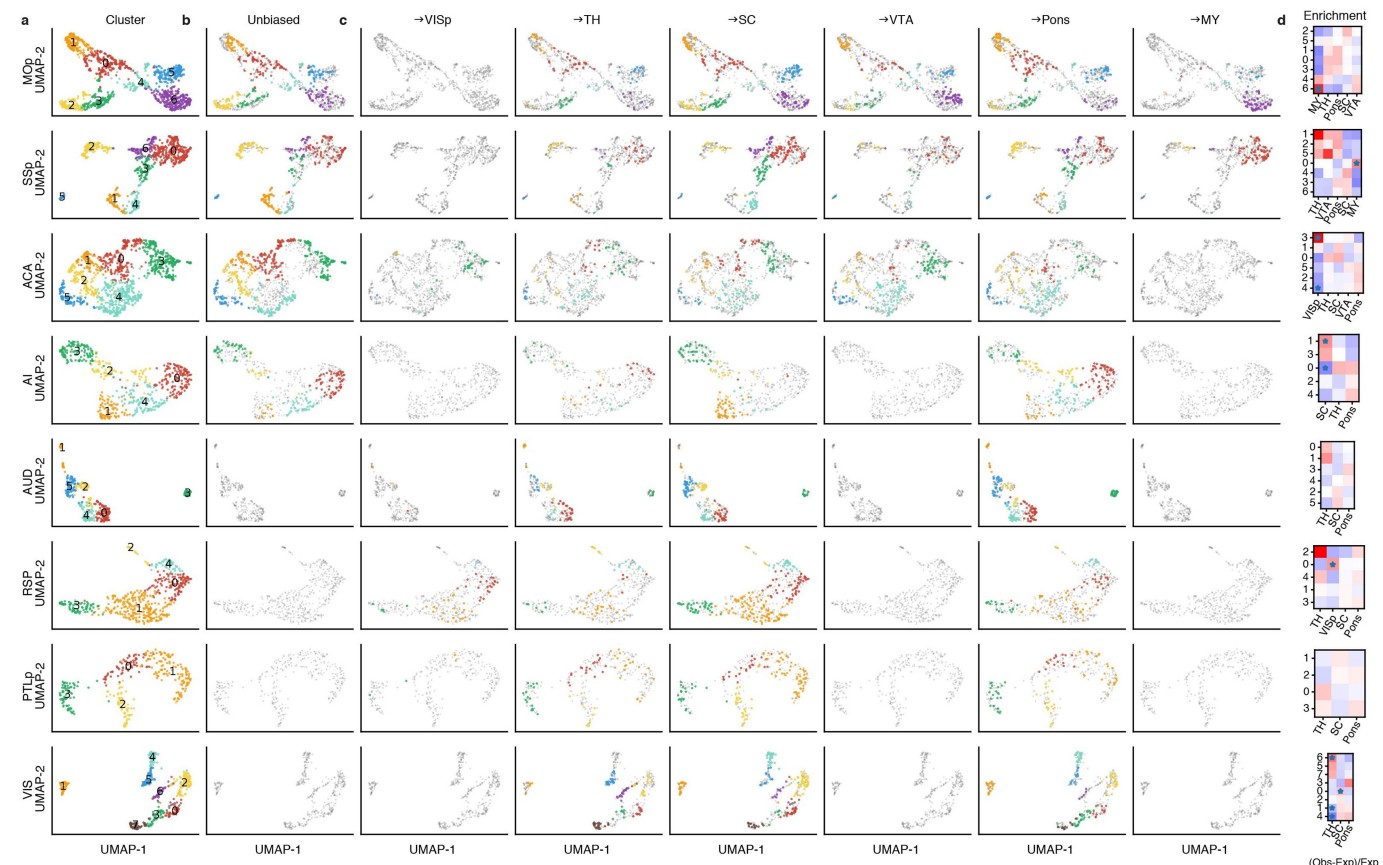

**Extended Data Fig. 7 | Enrichment of different projections in L5 ET clusters.**
**a**–**c**, *t*-SNE of L5 ET cells from each source coloured by clusters. The coloured cells are all cells (**a**), unbiased snmC-seq cells (**b**), and cells projecting to each target (**c**). Other cells were greyed. **d**, The enrichment of each projection in each L5 ET cluster in each source. *FDR < 0.05.

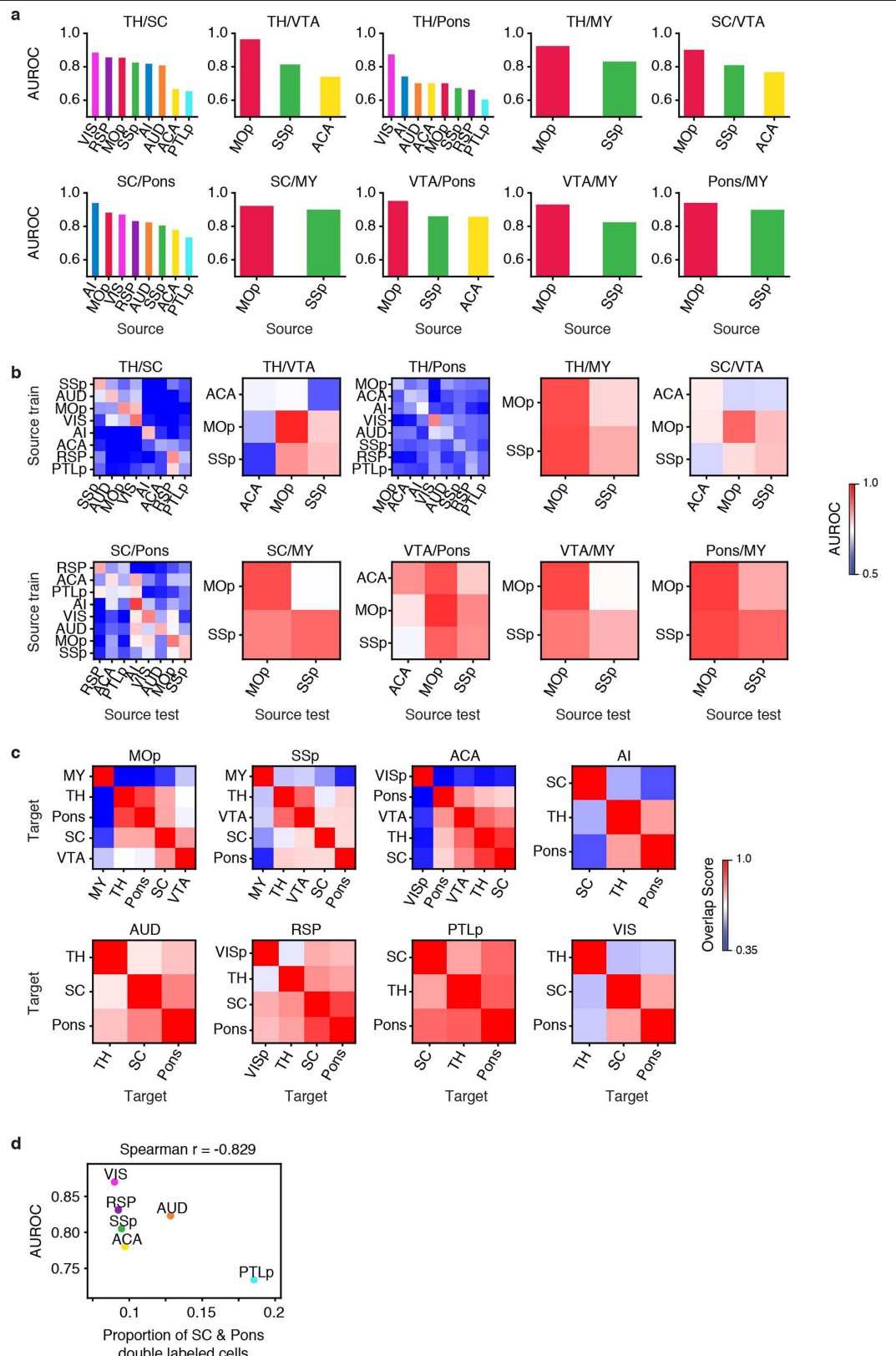

**Extended Data Fig. 8 | AUROC of ET target pairs within and across sources.** **a**, **b**, AUROC of models trained and tested in the same source (**a**) or models tested in all source regions after being trained in each one of them (**b**) using 100-kb bin mCH as features. Training and testing sets were randomly split. **c**, Overlap score between each pair of targets in each source. **d**, The proportion of double-labelled cells versus the AUROC score to distinguish superior colliculus versus pons neurons across sources.

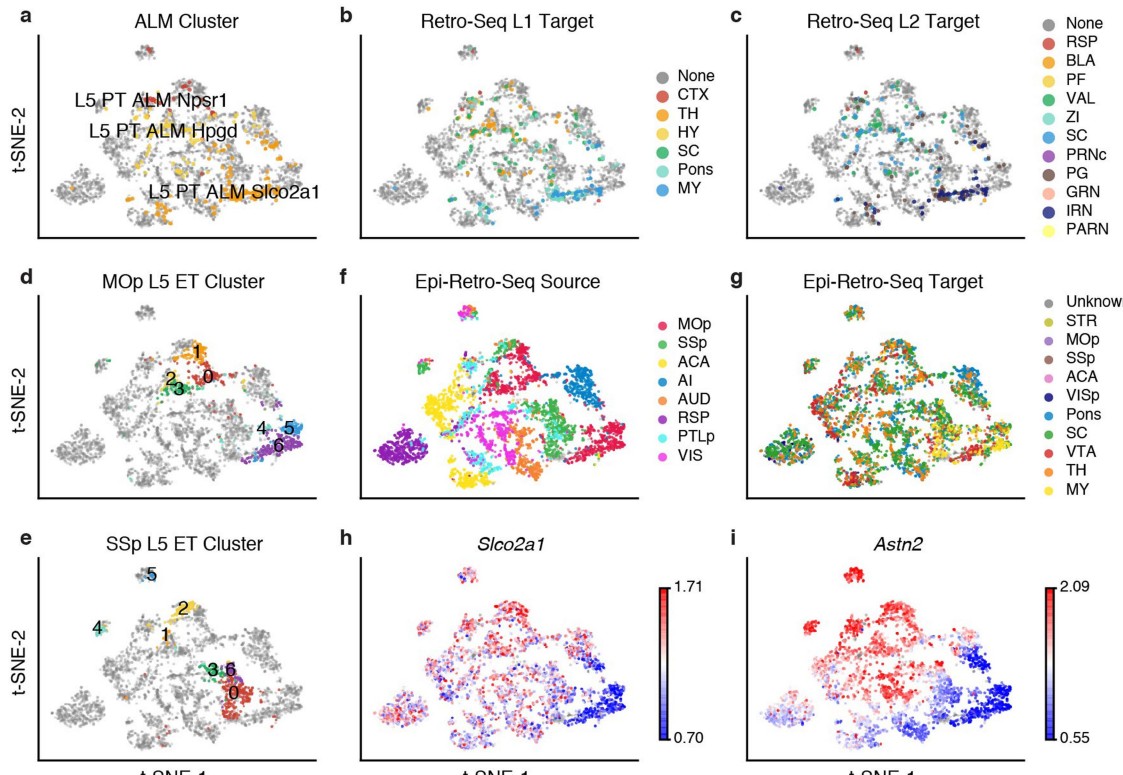

**Extended Data Fig. 9 | Integration of L5 ET cells from epi-retro-seq and retro-seq. a–c**, L5 ET ALM cells in SMART-Seq (*n* = 365) coloured by clusters (**a**), major target regions (**b**), and detailed target regions (**c**). Epi-retro-seq cells were greyed. **d–i**, L5 ET cells in epi-retro-seq from all source regions (*n* = 4,176) coloured by MOp clusters (**d**), SSp clusters (**e**), sources (**f**), targets (**g**), and gene body mCH of *Slco2a1* (**h**) and *Astn2* (**i**).

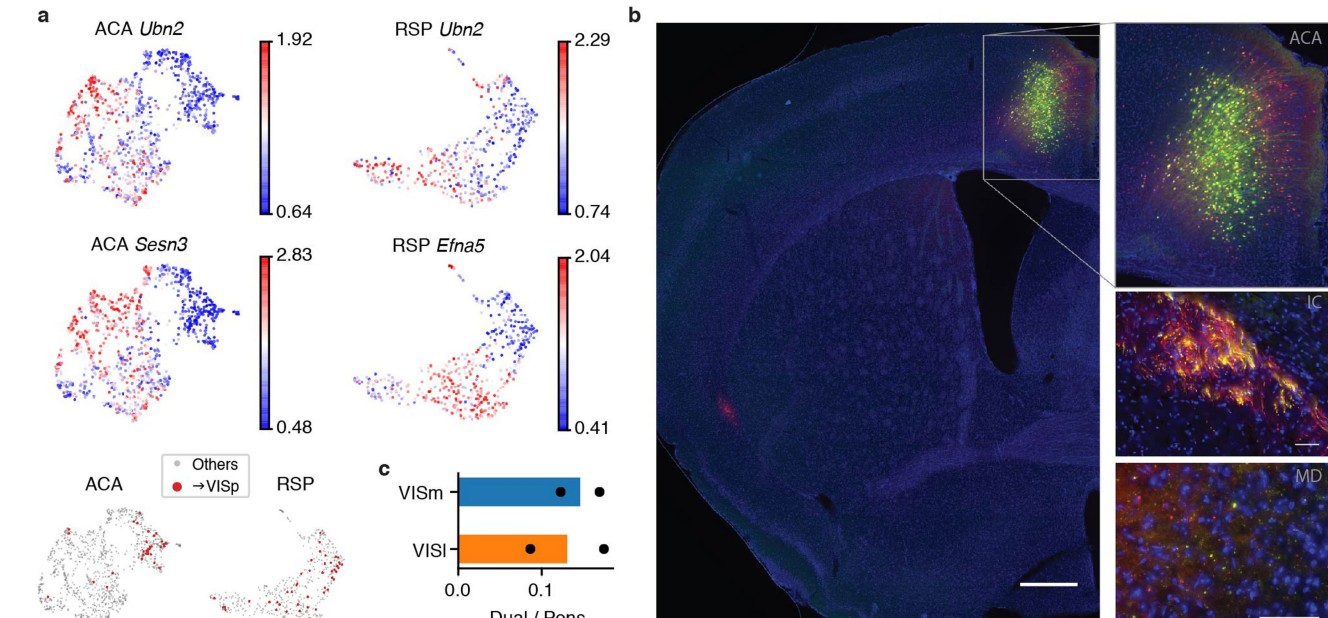

**Extended Data Fig. 10 | Validation of L5 ET + CC neurons. a**, UMAP of ACA ($n$ = 1,131) and RSP ($n$ = 516) L5 ET cells, coloured by gene body mCH of example genes. *Ubn2* shows hypomethylation in the cluster enriching neurons projecting to the VISp in both ACA and RSP, whereas *Sesn3* and *Efna5* are hypomethylated in the cluster only in ACA or RSP, respectively. VISp-projecting cells are shown in red at the bottom. **b**, By injecting AAV-retro-Cre in the VISp and AAV-FLEX-GFP in the ACA, the axon terminals of ACA–VISp neurons were also observed in the internal capsule (IC) and mediodorsal nucleus of thalamus (MD). Scale bars, 500 μm (left) and 50 μm (right in IC and MD). **c**, The proportion of double-labelled neurons that project to both VISp and pons, out of neurons projecting to pons in medial and lateral visual cortex (VISm and VISl, respectively). $n$ = 2 biological replicates are shown as individual points.

# nature research

# Reporting Summary

Nature Research wishes to improve the reproducibility of the work that we publish. This form provides structure for consistency and transparency in reporting. For further information on Nature Research policies, see Authors & Referees and the Editorial Policy Checklist.

## Statistics

For all statistical analyses, confirm that the following items are present in the figure legend, table legend, main text, or Methods section.

| n/a | Confirmed | |
|---|---|---|
| ☐ | ☒ | The exact sample size (*n*) for each experimental group/condition, given as a discrete number and unit of measurement |
| ☐ | ☒ | A statement on whether measurements were taken from distinct samples or whether the same sample was measured repeatedly |
| ☐ | ☒ | The statistical test(s) used AND whether they are one- or two-sided<br>*Only common tests should be described solely by name; describe more complex techniques in the Methods section.* |
| ☐ | ☒ | A description of all covariates tested |
| ☐ | ☒ | A description of any assumptions or corrections, such as tests of normality and adjustment for multiple comparisons |
| ☐ | ☒ | A full description of the statistical parameters including central tendency (e.g. means) or other basic estimates (e.g. regression coefficient) AND variation (e.g. standard deviation) or associated estimates of uncertainty (e.g. confidence intervals) |
| ☐ | ☒ | For null hypothesis testing, the test statistic (e.g. *F*, *t*, *r*) with confidence intervals, effect sizes, degrees of freedom and *P* value noted<br>*Give P values as exact values whenever suitable.* |
| ☒ | ☐ | For Bayesian analysis, information on the choice of priors and Markov chain Monte Carlo settings |
| ☒ | ☐ | For hierarchical and complex designs, identification of the appropriate level for tests and full reporting of outcomes |
| ☐ | ☒ | Estimates of effect sizes (e.g. Cohen's *d*, Pearson's *r*), indicating how they were calculated |

*Our web collection on statistics for biologists contains articles on many of the points above.*

## Software and code

Policy information about availability of computer code

| | |
|---|---|
| Data collection | BD Influx Sortware v1.2.0.142 (flow cytometry), Freedom EVOware v2.7 (library preparation), Illumina MiSeq control software v3.1.0.13 and NovaSeq 6000 control software v1.6.0/RTA v3.4.4 (sequencing), Olympus cellSens Dimension 1.8 (image acquisition) |
| Data analysis | FIJI distribution of ImageJ, Scikit-learn 0.20.3, Bedtools 2.27, Scanpy 1.6.0, fimo 5.0.2<br>cemba_data mapping pipeline is available at https://github.com/lhqing/cemba_data.git, including Cutadapt 1.18, Bismark 0.20.0, Bowtie2 2.3.5, Fastqc=0.11, Picard=2.18, Samtools=1.9, Htslib=1.9<br>scanorama: https://github.com/brianhie/scanorama.git<br>Other code are available on https://github.com/zhoujt1994/EpiRetroSeq2020.git |

For manuscripts utilizing custom algorithms or software that are central to the research but not yet described in published literature, software must be made available to editors/reviewers. We strongly encourage code deposition in a community repository (e.g. GitHub). See the Nature Research guidelines for submitting code & software for further information.

## Data

Policy information about availability of data

All manuscripts must include a data availability statement. This statement should provide the following information, where applicable:
- Accession codes, unique identifiers, or web links for publicly available datasets
- A list of figures that have associated raw data
- A description of any restrictions on data availability

The data analyzed in this study were produced through the Brain Initiative Cell Census Network (BICCN: RRID:SCR_015820) and deposited to NCBI GEO/SRA with accession number GSE150170 and the NEMO Archive (RRID:SCR_002001) under identifier nemo:dat-t2mznz0 accessible at https://assets.nemoarchive.org/dat-t2mznz0. The code for all of the analyses and the link to data browser can be found at https://github.com/zhoujt1994/EpiRetroSeq2020.git

# Field-specific reporting

Please select the one below that is the best fit for your research. If you are not sure, read the appropriate sections before making your selection.

☒ Life sciences  ☐ Behavioural & social sciences  ☐ Ecological, evolutionary & environmental sciences

For a reference copy of the document with all sections, see nature.com/documents/nr-reporting-summary-flat.pdf

# Life sciences study design

All studies must disclose on these points even when the disclosure is negative.

| | |
|---|---|
| Sample size | 384 nuclei from each projection (except the MOp→SSp projection from which 768 nuclei were assayed). The sample size allowed us to obtain high coverage methylomes for each projection, and confidently identify differentially methylated genes. |
| Data exclusions | Poor quality nuclei were excluded from clustering if they failed to meet the following pre-established quality control (QC) thresholds:<br>< 500,000 non-clonal reads<br>> 1% non-conversion rate |
| Replication | At least 2 male and 2 female mice were injected with AAV-retro-Cre for each projection target. Male and female samples were pooled separately for nuclei preparation. Nuclei collected from the male and female pool were used as biological replicates in the downstream analyses. Methylomes of cells from different replicates are highly similar (Fig. 1e). Results in Fig. 5f and Extended Data Fig. 10b are reproducible in three biological replicates. Results in Fig. 5h and Extended Data Fig. 10c are reproducible in two biological replicates, and each data point represents one replicate. |
| Randomization | Randomization is not applicable, since the cells collected are random by nature. |
| Blinding | Blinding is not applicable, since all data are collected from mice. |

# Reporting for specific materials, systems and methods

We require information from authors about some types of materials, experimental systems and methods used in many studies. Here, indicate whether each material, system or method listed is relevant to your study. If you are not sure if a list item applies to your research, read the appropriate section before selecting a response.

### Materials & experimental systems

| n/a | Involved in the study |
|---|---|
| ☐ | ☒ Antibodies |
| ☒ | ☐ Eukaryotic cell lines |
| ☒ | ☐ Palaeontology |
| ☐ | ☒ Animals and other organisms |
| ☒ | ☐ Human research participants |
| ☒ | ☐ Clinical data |

### Methods

| n/a | Involved in the study |
|---|---|
| ☒ | ☐ ChIP-seq |
| ☐ | ☒ Flow cytometry |
| ☒ | ☐ MRI-based neuroimaging |

## Antibodies

| | |
|---|---|
| Antibodies used | anti-GFP antibody, dilution: 1:500, Alexa Fluor 488 (Invitrogen, A-21311)<br>anti-NeuN antibody, dilution: 1:300,  EMD Millipore MAB377 conjugated with Alexa Fluor 647 (Invitrogen A20173) |
| Validation | All antibodies have been previously published for use in immunohistochemistry  and  flow cytometry experiments. Anti-GFP antibody has been validated in Kim et al. Neuron 2020 (PMID: 32396852). |

## Animals and other organisms

Policy information about studies involving animals; ARRIVE guidelines recommended for reporting animal research

| | |
|---|---|
| Laboratory animals | 42-49 day old adult male and female INTACT mice (R26R-CAG-loxp-stop-loxp-Sun1-sfGFP-Myc maintained on C57BL/6J background) were used for Epi-Retro-Seq experiments. Adult wildtype C57BL/6J mice were used for double-retrograde labeling experiments. Housing condition: Temperature: 21-23 C, relative humidity: 61-63%. |
| Wild animals | The study did not involve wild animals. |
| Field-collected samples | The study did not involve samples collected from the field. |

| | |
|---|---|
| Ethics oversight | All experimental procedures using live animals were approved by the Salk Institute Animal Care and Use Committee. |

Note that full information on the approval of the study protocol must also be provided in the manuscript.

# Flow Cytometry

## Plots

Confirm that:

☒ The axis labels state the marker and fluorochrome used (e.g. CD4-FITC).

☒ The axis scales are clearly visible. Include numbers along axes only for bottom left plot of group (a 'group' is an analysis of identical markers).

☒ All plots are contour plots with outliers or pseudocolor plots.

☒ A numerical value for number of cells or percentage (with statistics) is provided.

## Methodology

| | |
|---|---|
| Sample preparation | Manually dissected mouse brain samples were snap-frozen on dry ice and stored at -80 °C.  Prior to nuclei preparation, for each projection, samples from 2 males and 2 females were pooled separately as biological replicates. The frozen brain tissues were transferred to a pre-chiled 2-mL dounce homogenizer with 1 mL ice-cold NIM buffer (0.25M sucrose, 25mM KCl, 5mM MgCl2, 10mM Tris-HCl (pH7.4), 1mM DTT (Sigma 646563), 10µl of protease inhibitor (Sigma P8340)), with 0.1% Triton X-100 and 5µM Hoechst 33342 (Invitrogen H3570), and gently homogenized on ice with the pre-chilled pestle 10-15 times. The homogenate was transferred to pre-chilled microcentrifuge tubes and centrifuged at 1000 rcf for 8 min at 4 °C to pellet the nuclei. The pellet was resuspended in 1 mL ice-cold NIM buffer, and again centrifuged at 1000 rcf for 8 min at 4 °C. The pellet was then resuspended in 450 µL of ice-cold NSB buffer (0.25M sucrose, 5mM MgCl2, 10mM Tris-HCl (pH7.4), 1mM DTT, 9ul of Protease inhibitor), and filtered through 40µM cell strainer. The filtered nuclei suspension was incubated on ice for at least 30 minutes with 50µl of nuclease-free BSA for at least 10 minutes, then incubated with GFP antibody, Alexa Fluor 488 (Invitrogen, A-21311) and anti-NeuN antibody (EMD Millipore MAB377) conjugated with Alexa Fluor 647 (Invitrogen A20173). GFP+/NeuN+ single nuclei were isolated using fluorescence-activated nuclei sorting (FANS) on a BD Influx sorter with 100µm nozzle, and sorted into 384-well plates preloaded with 2µl of digestion buffer for snmC-seq215 (20mL digestion buffer consists of 10mL M-digestion buffer (2×, Zymo D5021-9), 1ml Proteinase K (20mg, Zymo D3001-2-20), 9mL water, and 10µL unmethylated lambda DNA (100pg/µL, Promega, D1521)). The collected plates were incubated at 50 °C for 20 minutes then stored at -20 °C. |
| Instrument | BD Influx |
| Software | BD Influx Sortware v1.2.0.142 |
| Cell population abundance | We sorted NeuN-positive and GFP-positive nuclei. |
| Gating strategy | Intact nuclei were first discriminated from debris by virtue of their bright DNA labeling (Hoechst Height signal) followed by light scattering profiles (Forward Scatter (FSC) Height vs Side Scatter (SSC) Height). Events with high Pulse Width measurements for FSC and SSC were then excluded as aggregates. Next, NeuN-AlexaFluor 647 labelled neuronal nuclei were selected ("*670/30 640" Height) from which GFP positive nuclei were sorted ("*530/40 488" Height). |

☒ Tick this box to confirm that a figure exemplifying the gating strategy is provided in the Supplementary Information.

