## [Peer Review File · Nature]

Manuscript Title: Epigenomic Diversity of Cortical Projection Neurons in the Mouse Brain

Reviewer Comments & Author Rebuttals**Reviewer Reports on the Initial Version:**

Referees' comments:

Referee #1 (Remarks to the Author):

The authors present heroic work integrating connectivity of cortical projection neurons with their epigenomic features (DNA methylation) across projection types and cortical areas at a single-cell level, a new approach they term Epi-Retro-Seq. They employ a range of computational tools to test the predictive power of genome-wide DNA-methylation signatures on the elucidation of diversity (i.e., similarities and differences) among cortical projection neurons across cortical areas, projection targets (intra- and extra-telencephalic targets), and presumptive cortical layers. They exemplify the power of single-cell high-resolution DNA methylation analysis by identifying potential regulatory cis-elements as well as key transcription factors characteristic of subclusters of L5-ET neurons (based on enriched TF motifs on the regulatory elements). They further illustrate that the combined analysis of differentially methylated regulatory elements and motif enrichment analysis of TF families with shared motifs might hold high potential for the identification of cluster-specific downstream gene targets. Finally, the authors report what is claimed to be a novel L5-ET neuron type located in RSP (and potentially in ACA) with dual projections to ET and IT targets.

The integrated analysis of anatomical information and epigenomic features of cortical projection neurons is novel, powerful, and promising. Even though this work does not provide any unprecedented generalizable principle regarding molecular and cellular diversity of cortical projection neurons, the approach and dataset almost certainly will be of substantial use for those working in the broad fields of molecular genetics, single-cell techniques, and molecular neuroscience, etc.. However, this manuscript needs further clarification and validation of the approach and results before publication in Nature.

Figure 1

The study likely suffers from a considerable degree of contaminant projection neurons that are not retrolabeled.

- 10-15% contamination by non-neuronal cells and inhibitory interneurons indicates that there is very likely at least a similar level of contamination by non-retrolabeled projection neurons. Indeed, since the proportion of excitatory to inhibitory neurons in the cortex is about 4:1 (Sahara et al., 2012), one can reasonably assume that about 12% of excitatory neurons in the study would be considered false positive. The authors need to clarify if this assumption is correct or not. In the very probable case that this is correct, the authors need to assess the potential impacts of such contamination on the statistical power of their source-target predictions done in the study.

- It is important to identify the source(s) of contamination, because, depending on the underlying technical or biological reason(s), the degree of projection neuron contamination would be better estimated (see below specific examples that suggest a higher level of contamination). The authors claim that FACS purification is potentially the source of contamination, but it seems unlikely to explain the extent of contamination given the clear separation of GFP+/NeuN+ nuclei from non-retrolabeled nuclei (Fig. 1a). It might be informative to directly test this possibility by immunolabeling unpurified and FACS-purified nuclei, preferably with an antibody against myc in addition to those against GFP and NeuN, and check the proportion of triple-positive nuclei.

- To what extent is the spread of the viral particles to neighboring cells (in both the source and target areas) contributing to the contamination? The authors need to provide evidence for the robustness of viral labeling. They should also test if the source areas have any non-neuronal or interneurons labeled by AAV-retro. It would be informative to compare the AAV-retro approach with non-viral labeling approaches, such as CTB-based retrograde labeling, to see if AAV-retro labeling works as expected.
- The stereotaxic coordinate for pons injection seems incorrect, as it seems too deep. The authors need to verify this coordinate, and ensure that it is referenced to standard zero points.
- To what extent could computational clustering or annotation contribute to the contamination? It is not clear how the authors performed annotation of 10 major clusters. Which cell types do these genes represent? Based on what criteria are these genes selected? How are they used for clustering? (ext. Fig. 2f)
- It is unclear how the authors sorted out the dual projecting neurons in the study. For example, how did the authors deal with neurons retrogradely labeled from the striatum, since all ET axons pass through the internal capsule, and since a significant fraction of them even have collaterals in the striatum? (Figure 1j). If an ET neuron with multiple projections is retrolabeled from only one of its targets, wouldn't this confound some of the similarity predictions performed among ET subclusters? (see below comments on Figure 4). The authors need to elaborate on this subject.
- Several examples of potential issues caused by contamination of non-retrolabeled neurons or by computational clustering artifacts include the following:
 - o How do the authors explain the supposed layer 2/3 neurons projecting to subcerebral targets (SC, VTA, Pons)? (Fig 1J)
 - o How do the authors explain that the proportion of TH-projecting neurons from L5-ET is almost three-fold higher than L6-CT (Fig 1J)?
 - o It is puzzling and contradictory to general knowledge in the field that half of the ET cluster presumably has high *Satb2* levels (Ext. Fig 1f). Is this reflected in the expression of *Satb2* by these cells?
 - o What do the authors think about the size of the L6-IT cluster in comparison to the rest of IT neurons or the L2-3 cluster? According to retro-labeling data from a large number of studies, only 5-10% of IT neurons are located in layer VI (Olavarria and Van Sluyters 1983; MacDonald et al. 2018). What is the potential underlying technical reason for the claimed higher representation of L6 IT neurons? What criteria did the authors use to ascribe cells as L6-IT neurons? Also, according to the data presented in Figure 1d, the L6-IT cluster has a smaller number of striatal-targeting neurons, and arguably higher VisP-targeting neurons. Are these findings in line with recent high throughput single-cell-profiling or circuit connectivity studies (e.g., Economo, 2018; Graybuck, 2020)?

Figure 2

- The predictive similarity/distinguishability analyses among subclusters of IT neurons with different connectivity patterns are valuable, novel, and interesting. However, the concerns raised above regarding multiple types of apparent and potential contamination and inaccurate clustering seem quite likely to confound the reported findings here.

Figure 3

- Identification of ET sub-clusters, and the potential regulatory elements and genes that mark each sub-cluster based on CG and CH methylation patterns, as well as TF binding motif analyses, is striking and valuable for the field. Preferably, validating these findings by standard approaches (ICC, ISH, and data from an online resource) would be very useful, and would importantly validate or refute and strengthen or weaken the manuscript.

- Gene ontology analysis doesn't provide substantially relevant information, and actually seems to weaken the findings. Related to this, three of the four terms reported in Fig 3e belong to biological pathways that are presumably/normally thought to be active during early neuronal development, which might confuse readers. Is it possible that methylation status in certain genes of adult cortical projection neurons reflect their past gene expression during early development rather than ongoing transcription? Explaining what the authors think might clarify this ambiguity.

Figure 4

- How are neurons with multiple collaterals to multiple ET targets taken into consideration in this analysis? These types of projection neurons are not at all uncommon. Is it possible or even extremely likely that some of the conclusions that the authors draw regarding similarity or distance of ET subpopulations are confounded by prevalent dual- or multi-projection neurons? The pons vs. SC comparison analysis needs to be revisited in case the stereotactic coordinates used in this study are incorrect.

- Why didn't the authors analyze AI or MOp, the top two areas showing the highest distinguishability (Ext. Fig. 8A) in the CTB retrograde labeling experiment that examines the proportion of dual-projection neurons targeting both pons and SC (Supplementary Table 7)? Is there any logical reason to exclude these areas from this analysis? Is it because there are no labeled cells in these areas in this experiment?

Figure 5

- The authors claim to identify a new L5-ET population in RSP that sends dual axons to VISp and ET targets, but the data shown in Fig 5 are not entirely convincing. This may simply reflect that RSP L5-ET neurons' main ET axon trajectory is going through the VISp area (see Allen Connectivity Atlas experiments 536920234 and 168164230), which can take up AAV-retro injected into VISp. Different approaches, such as rabies tracing or CTB dual-injections, are required to orthogonally investigate if the L5-ET-CC population in ACA and RSP indeed project to ET and cortical targets simultaneously.

- ACA is well-separated from VISp compared RSP; therefore, it might be more appropriate and presumably easier to investigate dual projection by L5-ET neurons.

- Also, in addition to several previous reports that the authors cite regarding the presence of dual projecting ET-neurons with intracortical projections, it is noteworthy that single-cell tracing data by the MouseLight project provides several such examples of ET neurons with intracortical collaterals (e.g. AA0764, AA0001, AA0011, AA0927, AA0250, AA0135, AA0576).

Referee #2 (Remarks to the Author):

Nature manuscript 2020-03-05043

Zhang et al use a combination of snmC-seq and retrograde labeling of neuronal projections (Epi-Retro-Seq) to characterize the DNA methylation signatures of cortical projection neurons from 8 cortical areas and 10 target regions. The authors provide the rationale that high resolution analysis

of cortical IT and L5-ET neurons has not been possible based on previous approaches, since these cell types are so closely related, and that Epi-Retro-Seq may help solve this issue.

The first 2 figures characterize the relationship between DNA methylation and source region, cortical layer, and target region. The authors observe that cortical layer is the primary determinant of methylation variability, followed by source region, then followed by target region. Many of the other analyses are expected and not very informative; e.g. some cell types are more closely related to each other than other types. The authors identify differentially methylated genes (DMGs) between neurons of different projection targets, and GO analysis shows that these genes are enriched in neuronal and synaptic pathways, which is unsurprising.

In figures 3 and 4, the authors analyze the L5-ET neurons (the most abundant cell type in their dataset) more closely, and reach similar conclusions as above: source region plays a bigger role in determining methylation differences than target region, and DMGs between subclusters are enriched in canonical neuronal functions. Gene regulatory network analysis predicted key transcription factors that might underlie these DMGs, based on the presence of TF motifs in regions with differential CG methylation that are linked to particular target genes. However, without any experimental validation of this approach, it is unclear whether anything new is learned.

Finally, in figure 5 the authors validate the finding in their Epi-Retro-Seq data that a subset of L5-ET neurons project to both IT and ET targets, which is the main biological finding in the paper.

This work is of high technical quality and represents largely an in-depth characterization of the epigenetic relationships between different cortical projection neurons. However, it is unclear what biological insights to our understanding of brain physiology are gained from these analyses. A notable biological insight is the identification of L5-ET neuron subtypes that have both inter-telencephalic (IT) and extra-telencephalic (ET) projections, but the functional implications are not further explored. Thus, because the scientific question being investigated, though of substantial significance to neuroscientists, is quite narrow and may be of lesser interest to a wider scientific readership. I wonder, therefore, whether this work is of sufficient general interest for the broader readership of Nature rather than for a more focused journal. This, however, is an editorial decision.

MAJOR COMMENTS:

1) In terms of approach and statistical analyses, I don't have any major concerns, although it is difficult to assess some of the computational methods without having significant personal expertise in applying these programs. The biggest concern with statistics (mentioned below) is that it appears that only ~37 neurons on average are assayed per sample, which seems very underpowered.

2) Many of the figures/figure panels are unnecessarily complicated and confusing, and don't always convey useful information. For example, in Figure 1D, it is almost impossible to get a sense of how much clustering there is by target region

- could the same plot be color coded with just two colors - one for cortical and one for subcortical targets? In addition, Extended Data Fig 3a is far too complicated and does not convey much useful information that is not already presented in other figure panels and/or discussed in the text. The authors present every conceivable comparison between source region, target region, and layer, but the vast majority of these comparisons do not seem informative.

3) The authors need to clarify the number of cells that were taken into account in their analysis per individual sample. It appears from the article that the entire study includes 320 samples for a total of 11,827 neurons analyzed. If this is the case, only an average of 37 neurons were analyzed per sample which could alter the clustering efficiency.

- 4) In Extended Data Fig.9, the authors compare Epi-Retro-Seq data with previously published Retro-seq data (scRNA-seq). This should be expanded to as much of the snmC-seq data set as possible to compare methylation with expression
- 5) Why are biological replicates not more related than by chance in Figure 1F? If replicates correspond to the same source regions from different animals, shouldn't they be correlated, if not as highly as neurons from the same source region in an individual animal?

MINOR COMMENTS

1) The construction of the Epi-Retro-Seq vector is not described in the paper (and not referenced as all references to the method are to previous papers of the authors on methylation analysis). This makes it difficult and cumbersome to follow the approach. As I understand the very superficially described approach as outlined in figure 1a: They inject the Cre virus into the "target" region and retrogradely label the neurons that innervate the target. Retrogradely would mean that the virus has to travel via the axon back to the corresponding cell body in the source region as indicated in the figure. The cell bodies in the source would become GFP positive if Cre gets into the nucleus and thus these neurons can be isolated indicating as having been innervated the target region (the target of these neurons). Is my understanding of the approach correct? If so, how efficient is the infection of axons and labeling of cell bodies? Also: they need to explain the rationale for transduction of cholera toxin subunit B / Alexa red. The approach needs to be better explained to facilitate understanding.

2) Is there concordance between differentially methylated genes (based on gene body CH methylation) and differential CG methylation at the promoters of these

genes? That analysis would provide a nice snapshot of how concordant these two inferences of transcriptional activity are

3) Since FANS is known to induce epigenetic artifacts, such as drastic upregulation of immediate early genes (Fos, Jun, etc), and this may involve artifactual changes in DNA methylation, are there any single-cell methylation data on unsorted mouse cortex that can be used to quantify the extent of such artifacts in the Epi-Retro-Seq data?

4) It would be more appropriate to use the phrase "DNA methylation diversity" and "DNA methylation differences" rather than "Epigenetic diversity" and "Epigenetic differences"

5) The resolution of Figure 2 should be improved.

6) Results should be described in the same order as they appear in the figures to facilitate the reading.

7) Line 1087, replace "Epi-seq" by "Retro-seq".

8) In line 390, replace "expression" with "methylation"

Author Rebuttals to Initial Comments:

Referees' comments:

Referee #1 (Remarks to the Author):

The authors present heroic work integrating connectivity of cortical projection neurons with their epigenomic features (DNA methylation) across projection types and cortical areas at a single-cell level, a new approach they term Epi-Retro-Seq. They employ a range of computational tools to test the predictive power of genome-wide DNA- methylation signatures on the elucidation of diversity (i.e., similarities and differences) among cortical projection neurons across cortical areas, projection targets (intra- and extra-telencephalic targets), and presumptive cortical layers.

They exemplify the power of single-cell high-resolution DNA methylation analysis by identifying potential regulatory cis-elements as well as key transcription factors characteristic of subclusters of L5-ET neurons (based on enriched TF motifs on the regulatory elements). They further illustrate that the combined analysis of differentially methylated regulatory elements and motif enrichment analysis of TF families with shared motifs might hold high potential for the identification of cluster-specific downstream gene targets. Finally, the authors report what is claimed to be a novel L5-ET neuron type located in RSP (and potentially in ACA) with dual projections to ET and IT targets.

The integrated analysis of anatomical information and epigenomic features of cortical projection neurons is novel, powerful, and promising. Even though this work does not provide any unprecedented generalizable principle regarding molecular and cellular diversity of cortical projection neurons, the approach and dataset almost certainly will be of substantial use for those working in the broad fields of molecular genetics, single-cell techniques, and molecular neuroscience, etc.. However, this manuscript needs further clarification and validation of the approach and results before publication in Nature.

We thank the reviewer for appreciating the significance and impact of our manuscript, and raising concerns that were very helpful to us in improving our analyses and presentation of the data.

We note here that a major concern that pervaded many of the reviewer's specific comments was the possibility of contamination. That is, neurons that were sampled might not have actually been projecting to the putative target region. Indeed, we presented data directly demonstrating that such contamination existed. But in the previous version of the manuscript, we chose to include all data in the main figures and leave the reader to try to assess the extent to which contamination might have affected the various samples. In response to the reviewer's concerns, we developed rigorous quantitative methods for assessing contamination that allowed us to distinguish between different mechanisms leading to contamination and also led us to exclude cases with unacceptable levels of contamination. As a result nearly all data were reanalyzed and new figures generated after application of the exclusion criteria. While there were not any changes to the major conclusions of the manuscript, there were many smaller changes. We have highlighted all figures and panels that have changed by using red text in the figure legends. We have also highlighted all changes to the text in red. Below, we provide more detailed descriptions of these and other changes in the context of the specific comments.

Figure 1

The study likely suffers from a considerable degree of contaminant projection

neurons that are not retrolabeled.

- 10-15% contamination by non-neuronal cells and inhibitory interneurons indicates that there is very likely at least a similar level of contamination by non-retrolabeled projection neurons. Indeed, since the proportion of excitatory to inhibitory neurons in the cortex is about 4:1 (Sahara et al., 2012), one can reasonably assume that about 12% of excitatory neurons in the study would be considered false positive. The authors need to clarify if this assumption is correct or not. In the very probable case that this is correct, the authors need to assess the potential impacts of such contamination on the statistical power of their source-target predictions done in the study.

We appreciate the reviewer's suggestion to assess contamination based on the presence of cell types, such as inhibitory neurons, that are not expected to be projecting to the intended targets. We have taken this basic approach to develop new exclusion criteria, but note that we can improve the power of the analyses by also considering other known off-target cell types. These off target cell types vary depending on the intended target. As detailed below, we implemented analyses to identify experiments in which samples were likely contaminated by unacceptable numbers of neurons that did not project to the intended target. Neurons from these samples were designated as having "unknown" projections.

Generally, there are two possible mechanisms that can lead to contamination. 1) Labeling errors. Cells that do not project to the intended target become labeled by AAVretro-Cre and express GFP. 2) Sorting errors. We identified two types of sorting errors. 2a) FANS gating is set poorly for NeuN (resulting in contamination by glia), or GFP (resulting in contamination by off-target cells). 2b) FANS gates are acceptable but the number of gate- positive neurons in the sample is very small such that the few "wrong" cells at the edges of the gate constitute a high proportion of the sorted sample. Labeling errors depend on injection and source interactions while FANS errors vary between sorting runs. We have now conducted a rigorous analysis of contamination in each of the experiments that were conducted.

The approach that we took was to assess the data from every nuclei sorting case independently to evaluate their levels of contamination. Because each sorting case corresponds to a unique source/target pair as well as a unique sort, both types of errors can be assessed at this level. For each sort, we quantified the proportions of neurons that were present in expected on-target clusters versus known off-target clusters and compared that to the proportions of neurons expected from unbiased cortical samples (Liu *et al.*, 2020). Samples were excluded if they did not meet on-target fold enrichment criteria and have statistically significant enrichment. The expected enrichment values are different for ET targets than for IT (including striatum) targets, so the enrichment thresholds vary slightly depending on the target

area.

For ET targets, all correctly labeled and sorted neurons should be in either the L5-ET or the L6-CT cluster. There should be very few inhibitory neurons (a very small number do make projections to ET targets) and no IT neurons. Therefore, for each ET target sorting case we calculated the ratio of L5-ET cluster neurons to IT+inhibitory neurons. These ratios were compared to the expected ratio and cases in which there was not at least 5-fold enrichment above the expected values, or in which there was not statistically significant enrichment (FDR<0.01, exact binomial test of goodness-of-fit and Benjamini-Hochberg procedure) were excluded. This eliminated 7 out of 101 ET target sorts (285 out of 5364 cells, see Fig. R1b-left below). Close inspection of each of these cases revealed that 1 was a 2b error apparent from a very small sample (only 8 sorted neurons total), detected by the statistical criterion and inspection of sorting gates. The other 6 cases could be attributed to error 1. All 6 cases corresponded to injections of AAVretro into either the TH or VTA, and contaminated sources were cortical regions through which the injection pipettes extended. (See example results from a AI→TH sorting case shown in Fig. R1a-left. This case contained many IT and inhibitory neurons. We also include data from the other AI→TH replicate in which the injection did not result in significant contamination (Fig. R1a, right).)

We were aware of this error source from a less rigorous inspection of the cell type clusters observed following TH and VTA injections (old Fig.1j, new Extended Data Fig. 2h). The new analyses allowed us to precisely identify the experiments that led to the contamination and to exclude them. It is important to note that the 5-fold enrichment still leaves some neurons from “unexpected” targets in the included sample (Fig. 1j, Extended Data Fig. 3), but these are computationally eliminated from analyses of ET cells, as are any glial cells. The 5-fold criterion assures that the neurons within the L5-ET cluster are highly enriched for those that project to the intended target versus some other L5-ET target. Finally, it is noteworthy that, as now stated in the manuscript text, this mechanism of contamination is apparent in previously published retro-seq data (Tasic *et al.*, 2018) where no effort was made to identify or correct that contamination. Overall, the new analysis had little effect on our overall results because other dissected slices and biological replicates from the same target injection cases yielded considerably more sorted neurons.

We applied similar methods to the IT and striatum target cases, but here the on-target neurons are IT clusters and the off-target are inhibitory and L6-CT clusters. L5-ET projections to striatum are not considered because they are a small proportion and have a negligible influence on the computations. Here, unlike for the ET targets in which the expected proportion of on-target cells is very small, the proportion of

expected on target neurons is expected to be much higher than the off-target. Thus the calculated ratios are noisier and require more lenient fold-change criteria (3-fold) to avoid elimination of data that were not clearly contaminated. The same statistical criterion was applied. These criteria led to exclusion of 30 out of 115 sorting cases (603 out of 5043 neurons, Fig. R1b- right). Fortunately, all of our IT targets are at the surface of the brain and therefore injection pipettes do not pass through any of the source regions, obviating the possibility of type 1 errors. Closer examination of each sort revealed both 2a and 2b errors.

Fig. R1. Removing FANS runs with high contamination. a, The distribution of cells across clusters from two different FANS runs to select AI→Pons neurons. **b**, The scatter plots for filtering FANS runs with high contamination. Each dot represents a single run, and the size of dot represents the number of on-target cells selected by the run.

We have updated the Methods section to describe these new analyses and inclusion criteria (Line 801-836). We have also added a brief description of contamination sources and methods used to eliminate them within the main text (Line 125-135). Fig. R1b has been incorporated into Extended Data Fig. 2i.

As noted above, we have also reanalyzed all of the data related to projection targets and updated all figures and descriptions throughout the manuscript that were affected by this change.

- It is important to identify the source(s) of contamination, because, depending on the underlying technical or biological reason(s), the degree of projection neuron contamination would be better estimated (see below specific examples that suggest a higher level of contamination). The authors claim that FACS purification is potentially the source of contamination, but it seems unlikely to explain the extent of contamination given the clear separation of GFP+/NeuN+ nuclei from non-retrolabeled nuclei (Fig. 1a). It might be informative to directly test this possibility by immunolabeling unpurified and FACS- purified nuclei, preferably with an antibody against myc in addition to those against GFP and NeuN, and check the proportion of triple-positive nuclei.

Please see the consideration of these factors in the description above. As noted

there, we can attribute errors to known sources and we have identified and eliminated affected data. Note that Fig 1a is only one example out of 216 FACS sorting runs. The reviewer is correct that FACS errors do not account for all of the contamination. We appreciate the reviewer's suggestion for further tests. We have conducted experiments similar to those suggested using anti-GFP antibody followed by visual inspection, which verified that when the FACS gates were set correctly that the GFP+ nuclei were truly GFP+ (Kim *et al.*, 2020). However, the successful sorting in such a control experiment does not eliminate the possibility of sorting errors from other samples in which the gates are set independently. In addition, this does not reveal cases of contamination from type 1 (labeling) errors as described above. Instead, we used the new computational analysis described above to address this point.

- To what extent is the spread of the viral particles to neighboring cells (in both the source and target areas) contributing to the contamination? The authors need to provide evidence for the robustness of viral labeling. They should also test if the source areas have any non-neuronal or interneurons labeled by AAV-retro. It would be informative to compare the AAV-retro approach with non-viral labeling approaches, such as CTB-based retrograde labeling, to see if AAV-retro labeling works as expected.

This question is also addressed above. As suggested, we have analyzed the presence of unexpected cell types (including interneurons) that could have resulted from spread of viral particles to source areas adjacent to the target region. We did not find contamination that could be attributed to such spread, as expected from the fact that all of our dissected and sorted sources are distant from injection sites. In contrast, there was contamination from spread of the virus along the injection pipette track. Comparisons between AAVretro and CTB can be found in the original publication describing retrograde infection with this AAV serotype (Tervo *et al.*, 2016). With respect to robustness (efficiency of labeling), this is not a factor in our experimental design; we are sampling from a subset of neurons that project to a target area, so labeling more cells would not change the results.

- The stereotaxic coordinate for pons injection seems incorrect, as it seems too deep. The authors need to verify this coordinate and ensure that it is referenced to standard zero points.

The coordinates for pons vary between atlases, largely because very small shifts in AP position can result in large differences in depth. Our coordinates do correspond to the pons for the Allen Atlas (<https://mouse.brain-map.org/experiment/siv?id=100142143>, Coronal Level 89).

- To what extent could computational clustering or annotation contribute to the

contamination? It is not clear how the authors performed annotation of 10 major clusters. Which cell types do these genes represent? Based on what criteria are these genes selected? How are they used for clustering? (ext. Fig. 2f)

Clustering of single cells based on their methylome robustly identifies the major cell types with excellent correspondence to single-cell transcriptomic approaches (Luo *et al.*, 2017, 2019; Yao *et al.*, 2020a). For the major cell classes (ET, IT, inhibitory, L6-CT) accuracy is likely near 100% and thus the estimation of contamination based on ratios of on-target to off-target cells provides a highly accurate assessment. As described in the methods, we used genome-wide bin level information for clustering, rather than specific marker genes, to ensure the clusters captured the methylation features across the whole genome. The well-known marker genes in Extended Data Fig. 2f were only used to annotate which major cell type the clusters correspond to. Specifically, *Cux2*⁺ *Rorb*⁻ (hypo-methylation in *Cux2* gene body and hyper-methylation in *Rorb* gene body) was annotated as L2/3-IT; *Cux2*⁺ *Rorb*⁺ was annotated as L4-IT; *Cux2*⁻ *Rorb*⁺ and *Deptor*⁺ were annotated as L5-IT; *Sulf1*⁺ and *Sulf2*⁺ *Deptor*⁻ were annotated as L6-IT; *Vat1l*⁺ was annotated as L5-ET; *Foxp2*⁺ was annotated as L6-CT; *Tle4*⁺ *Foxp2*⁻ was annotated as L6b; *Tshz2*⁺ was annotated as NP; *B3gat2*⁺ was annotated as CLA; *Slc6a1*⁺ was annotated as Inh. The clusters with low global mCH level were annotated as non-neural cells, which were further confirmed by hyper-methylation of *Mef2c*. We have added the information of cluster annotation in the Methods (Line 792-798).

It is unclear how the authors sorted out the dual projecting neurons in the study. For example, how did the authors deal with neurons retrogradely labeled from the striatum, since all ET axons pass through the internal capsule, and since a significant fraction of them even have collaterals in the striatum? (Figure 1j). If an ET neuron with multiple projections is retrolabeled from only one of its targets, wouldn't this confound some of the similarity predictions performed among ET subclusters? (see below comments on Figure 4). The authors need to elaborate on this subject.

Our study does not sort out dual projecting neurons. This is inherent to our experimental design, that we only study one target of the neurons while not considering the other possible projection targets of the neurons.

Thus, the aim of our study is to subsample a certain number of neurons projecting to each target and analyze their methylation differences without conditioning on the other targets of the neurons. This also helped us to enrich for the less represented projections within each source (e.g. ET neurons). The full projection pattern of neurons would need to be analyzed with more complex tracing strategies in the future. For example, the follow-up studies that we conducted to confirm the presence of L5-ET+CC neurons.

We acknowledged the widespread existence of neurons projecting to multiple targets simultaneously. As discussed in methods (Line 878 to 888), the prediction performance is affected by 1) proportion of single neurons projecting to both targets, 2) neurons sharing the same methylation signatures but projecting differently, 3) the variation across replicates, 4) the potential contamination, 5) the sample size, and 6) the power of the model. We have validated in Fig. 4k that the prediction performances are indeed correlated with the proportion of neurons making dual projections.

- Several examples of potential issues caused by contamination of non-retrolabeled neurons or by computational clustering artifacts include the following:

o How do the authors explain the supposed layer 2/3 neurons projecting to subcerebral targets (SC, VTA, Pons)? (Fig 1J)

This has been addressed in detail above. These are in fact L2/3 neurons that were labeled due to passage of the injection pipette through the source region on the way to the target region. Our new exclusion criteria assure the elimination of highly contaminated samples and that the L5-ET cluster is substantially enriched with neurons projecting to the intended target. The revised Fig. 1j shows that there are still small numbers of off-target neurons, as expected from our exclusion criteria. Note that an unbiased sample would contain 10- fold more IT neurons than ET neurons, so even though our ET-projecting samples are highly enriched for L5-ET cells that project to the target region, there remain some IT cells from contamination. As can be seen in Fig. 1J, for most targets (SC, VTA, pons, MY) there are at least 3 to 5-fold more ET than IT cells indicating an enrichment of 30 to 50- fold. Thus, only about 2-3% of the L5-ET neurons from these cases are likely contaminated cells that did not project to the intended target. For the TH target there is somewhat more contamination, but enrichment is still 15-fold; thus up to 6% of the L5-ET cells are off-target. Please note that the presence of IT or inhibitory cells in these samples does not affect subsequent analyses that are restricted to the L5-ET cluster; IT and inhibitory cells are computationally removed in these analyses.

o How do the authors explain that the proportion of TH-projecting neurons from L5-ET is almost three-fold higher than L6-CT (Fig 1J)?

There are two factors that affect the ratio of L5-ET/L6-CT cells in our sample. First, the actual ratio of cells varies depending on the thalamic target nucleus and the cortical area sampled. Second, the efficiency of uptake of AAVretro is much greater for L5-ET cells than for L6-CT cells. Conversely, CTB does not efficiently label L5-ET cells. In our unpublished studies with injections of AAVretro or CTB into thalamus we often see AAV labeling restricted to L5 and CTB labeling largely restricted to L6

of VISp (Fig. R2). Our results (sampling bias for L5-ET cells) would not be expected from any potential non-specific contaminating mechanism because the actual numbers of L6-CT cells is >2-fold greater than the L5-ET cells in the cortical regions we sampled. This sampling bias does not affect any of our conclusions.

Fig. R2. Comparison of laminar retrograde cortico-thalamic labeling patterns for CTB and AAVretro. Left panel shows that CTB retrograde labeling of neurons in VISp following injections in pulvinar nucleus of thalamus results in labeling predominantly in L6 and very little L5 label (arrows). Right panel shows that nearly all retrogradely labeled cortical neurons are in L5 following similar injections of AAVretro.

o It is puzzling and contradictory to general knowledge in the field that half of the ET cluster presumably has high *Satb2* levels (Ext. Fig 1f). Is this reflected in the expression of *Satb2* by these cells?

Satb2 is a marker of excitatory cells in the isocortex. We actually detect *Satb2* in nearly all of the L5-ET cells, but there are differences in the CH methylation levels within the L5- ET cluster. Note that the different plots in Extended Data Fig. 2f use different CH methylation levels in relation to the chosen color scale. Here we provide a rescaled plot showing that *Satb2* is hypomethylated in all the L5-ET cells and hypermethylated in the inhibitory cells (Fig. R3). We have not changed the color scale in the manuscript because we feel that the color scale is more informative if the full range of methylation levels is highlighted. The variation in mCH levels corresponds

to similar variation of expression levels of *Satb2* in L5-ET as observed in scRNA-seq analysis (Yao *et al.*, 2020b).

Fig. R3. Gene body mCH of *Satb2* in single cells (n=13,414).

o What do the authors think about the size of the L6-IT cluster in comparison to the rest of IT neurons or the L2-3 cluster? According to retro-labeling data from a large number of studies, only 5-10% of IT neurons are located in layer VI (Olavarria and Van Sluyters 1983; MacDonald *et al.* 2018). What is the potential underlying technical reason for the claimed higher representation of L6 IT neurons? What criteria did the authors use to ascribe cells as L6-IT neurons? Also, according to the data presented in Figure 1d, the L6-IT cluster has a smaller number of striatal-targeting neurons, and arguably higher VisP-targeting neurons. Are these findings in line with recent high throughput single-cell- profiling or circuit connectivity studies (e.g., Economo, 2018; Graybuck, 2020)?

Every method used that might reveal the proportion of IT cells that are in L6 is subject to sampling biases. In addition, the proportion varies depending on the source and target cortical areas. This can clearly be seen in our data (Extended Data Fig. 3a) as well as in data that are published or available online. The papers cited by the reviewer are for callosal projections in the rat and mouse cortex. Callosal projections are known to be biased toward L2/3 neurons. Therefore the numbers that we will observe are expected to be different because we have sampled very different projections in different species. A more representative estimate of the actual proportions is provided by Oberlander *et al.* 2012 (Oberlander *et al.*, 2012). They report only 1.6-fold more L2/3 than L6-IT cells in the rat barrel cortex. Our unbiased sample of cortical neurons (Liu *et al.*, 2020) has only 2-fold more L2/3-IT neurons than L6-IT in the mouse cortex. These numbers are similar to the range of values seen in the various cortico-cortical projections that we sampled (Fig. 1j and Extended Data Fig. 3a, the average ratio across our samples is 1.5).

Graybuck *et al.* 2020 does not provide data relevant to this issue, as it is a study testing the cells in which particular enhancers drive expression. Similarly, Economo *et al.* is not relevant as that is a study of gene expression in different types of L5-ET cells.

Figure 2

- The predictive similarity/distinguishability analyses among subclusters of IT neurons with different connectivity patterns are valuable, novel, and interesting. However, the concerns raised above regarding multiple types of apparent and

potential contamination and inaccurate clustering seem quite likely to confound the reported findings here.

We have addressed this issue in response to the comments above. There is also a discussion of this found in the methods section of the manuscript (Line 878-905). As noted above, we have now established rigorous exclusion criteria and analyses that assure that neurons in each sample are highly enriched for their intended target. The conclusions of distinguishability variation across source and targets remain the same, as well as the shared molecular distinction between projections across sources and layers. As noted above, the clustering and subsequent annotation are accurate.

Figure 3

- Identification of ET sub-clusters, and the potential regulatory elements and genes that mark each sub-cluster based on CG and CH methylation patterns, as well as TF binding motif analyses, is striking and valuable for the field. Preferably, validating these findings by standard approaches (ICC, ISH, and data from an online resource) would be very useful, and would importantly validate or refute and strengthen or weaken the manuscript.

We agree that it would be nice to be able to independently validate these predictions, however, we do not think that the suggested analyses would definitively address this question. The ET subclusters are distinguished based on their gene expression and/or methylation. They are not expected to be distinguishable, for example, based on their spatial distribution during an IHC or ISH experiment. Therefore it is not possible to correlate IHC or ISH with the same clusters. As we have pointed out in this manuscript, these data can be used in the future to predict cis-regulatory elements that might selectively bias gene expression to these genetically-defined clusters. Such future experiments would then make it possible to further characterize the cells whose expression is regulated by those new tools.

- Gene ontology analysis doesn't provide substantially relevant information, and actually seems to weaken the findings. Related to this, three of the four terms reported in Fig 3e belong to biological pathways that are presumably/normally thought to be active during early neuronal development, which might confuse readers. Is it possible that methylation status in certain genes of adult cortical projection neurons reflect their past gene expression during early development rather than ongoing transcription? Explaining what the authors think might clarify this ambiguity.

We agree that gene ontology names are simply a reflection of nomenclature that has been created in an effort to summarize many complex relationships between

genes and their function. We have presented these results using standardized methods and nomenclatures. Many different GO terms contain overlapping genes. The genes under the terms related to development are not necessarily repressed in the adult, and many of them are consistently expressed after cell-type specification. The full list of enriched GO terms was included in Supplementary Table 4, among which many GO terms were related to neuronal functions but not development. Now we have also added cautionary statements in Methods (Line 962-965) to clarify that we do not expect that these analyses can be used to directly infer how a particular gene contributes to neuronal function in a specific context. Accordingly, we do not feel that there is a need for identification of any other factors to resolve this apparent ambiguity. We do not feel that presentation of this analysis in any way weakens our findings, provided that the consumer is aware of the limitations of gene ontology nomenclature. We agree that adult methylation status can be related to early development, but we do not feel that we can make any definitive statements about the implications of these gene ontology findings with respect to development.

Figure 4

How are neurons with multiple collaterals to multiple ET targets taken into consideration in this analysis? These types of projection neurons are not at all uncommon. Is it possible or even extremely likely that some of the conclusions that the authors draw regarding similarity or distance of ET subpopulations are confounded by prevalent dual- or multi- projection neurons? The pons vs. SC comparison analysis needs to be revisited in case the stereotactic coordinates used in this study are incorrect.

We have carefully considered this issue in all of our analyses and conclusions. For example, most L5-ET neurons project to many different targets and the lack of clear epigenetic separation between most of these targets is expected from that fact. For example, many pons-projecting neurons are also SC-projecting neurons. We have further shown that the proportion of double labeling of these targets is correlated to the degree of epigenetic similarity in the various cortical source areas sampled. This analysis specifically addresses multiple projections.

As addressed in the previous question, the stereotactic coordinates used are correct.

- Why didn't the authors analyze AI or MOp, the top two areas showing the highest distinguishability (Ext. Fig. 8A) in the CTB retrograde labeling experiment that examines the proportion of dual-projection neurons targeting both pons and SC (Supplementary Table 7)? Is there any logical reason to exclude these areas from this analysis? Is it because there are no labeled cells in these areas in this experiment?

The reviewer has correctly inferred that in some cases there were not labeled cells from injections either in AI or MOp, as shown in Supplementary Table 7. In addition, as stated in our methods, we implemented additional inclusion criteria. These criteria required that labeled cells from the two injections must be intermingled in the same region: “Therefore, some cortical areas in which there was minimal or no overlap are not included.” We have revised supplementary Table 7 to include the raw data and to describe the exclusion criteria. We believe that these data actually support the expectation that the SC- versus pons-projecting cells would be most distinct within MOp and AI. But we have restricted our analysis according to the criteria because the lack of double labeling from non- overlapping populations could theoretically be related to the topographic organization of connections rather than a lack of dual projections.

Figure 5

The authors claim to identify a new L5-ET population in RSP that sends dual axons to VISp and ET targets, but the data shown in Fig 5 are not entirely convincing. This may simply reflect that RSP L5-ET neurons’ main ET axon trajectory is going through the VISp area (see Allen Connectivity Atlas experiments 536920234 and 168164230), which can take up AAV-retro injected into VISp. Different approaches, such as rabies tracing or CTB dual-injections, are required to orthogonally investigate if the L5-ET-CC population in ACA and RSP indeed project to ET and cortical targets simultaneously.

We have conducted additional anatomical experiments, as suggested by the reviewer, and all of the results add further confirmation of the presence of L5-ET+CC cells in both ACA and RSP, as well as additional cortical areas that were analyzed. We have also closely evaluated available online data and found, as described below, that ET-projecting cells do not have an axon trajectory through VISp.

We now present, as a new panel in Fig. 5, an analysis of double-labeled cells in RSP and ACA following injections of CTB into pons and VISp (Fig. R4a). A substantial proportion of the pons-projecting cells (a subset of L5-ET cells) are double-labeled by the VISp injections. There are about 25% double-labeled cells in RSP and 10% in ACA indicating that these cells are not uncommon (Fig. R4b). We also conducted the AAVretro/Cre- dependent GFP experiment for ACA (same as previously reported for RSP) and again found that the VISp-projecting neurons have collaterals in sub-cerebral targets (Fig. R4c). These results are shown in the revised manuscript Fig. 5g, h and Extended Data Fig. 10b, c.

Fig. R4. Validation of L5-ET+CC neurons. **a**, The diagram of experiment labeling neurons making dual projections to both VISp and pons. **b**, The proportion of double labeled neurons out of neurons projecting to pons in RSP, ACA, VISm, and VISl. **c**, By injecting AAVretro-Cre in VISp and AAV-FLEX-GFP in ACA, the axon terminals of ACA→VISp neurons were also observed in internal capsule (IC) and mediodorsal nucleus of thalamus (MD).

We thank the reviewer for pointing us to the cases available in the Allen Mouse Connectivity atlas. These data actually provide independent validation of our findings. Furthermore, they do not show that L5-ET cell axons projecting to subcortical structures pass under the primary visual cortex. Case #168164230 corresponds to a Cre-dependent AAV injection into RSP of an Rbp4-Cre mouse. This mouse line expresses Cre in both L5-ET and L5-IT neurons, as well as some L6-IT neurons (Fig. 2 in (Tasic *et al.*, 2016)). Thus, the labeled cells include L5-IT cells making projections to VISp as well as other neighboring areas.

A better assessment of the trajectory taken by L5-ET axons targeting sub-cerebral structures is available from single neuron reconstructions of Mouselight data (<http://ml-neuronbrowser.janelia.org/>) which show that projections of L5-ET cells to sub-cerebral structures do not pass under VISp. For example, Neuron AA1045 (Fig. R5) is an RSP L5-ET cell that can clearly be seen to take a trajectory into the pyramidal tract (as expected) and forms collaterals in thalamus, pons and SC. The axons are all quite distant from and do not pass under VISp.

[Figure R5 Redacted (DOI: 10.25378/janelia.7822310)]

Fig. R5. Reconstruction of the axonal projection of a L5-ET neuron in RSP. Note that axons target TH, SC and pons and do not pass near VISp.

The reviewer refers to Allen Institute Case #536920234. This is actually an experiment that is nearly identical to the experiment that we conducted to demonstrate the dual projections from RSP to both VISp and sub-cerebral structures. Retrograde-infecting CAV-Cre was injected into VISp and AAV with Cre-dependent GFP was injected into RSP. Labeling was found in the same subcerebral structures that we found. This experiment, therefore, provides independent validation of our results. We thank the reviewer for pointing this out and we have added a citation to this case in our revised manuscript.

- ACA is well-separated from VISp compared RSP; therefore, it might be more appropriate and presumably easier to investigate dual projection by L5-ET neurons.

As suggested by the reviewer and described above we have conducted additional anatomical studies validating the presence of L5-ET+CC neurons in ACA (Fig. R4c). Altogether, our results now show that this cell type is not unique to RSP. We show that it is also present in ACA and in visual areas lateral and medial to VISp (Fig. R4b). As we noted in the original manuscript, previous studies have also found such cells in the motor cortex. The additional data that we have now collected from dual retrograde labeling, as well as studies of projections from ACA, show that this is a widespread phenomenon. These results are shown in the revised manuscript Extended Data Fig. 10b, c.

- Also, in addition to several previous reports that the authors cite regarding the presence of dual projecting ET-neurons with intracortical projections, it is noteworthy that single-cell tracing data by the MouseLight project provides several such examples of ET neurons with intracortical collaterals (e.g. AA0764, AA0001, AA0011, AA0927, AA0250, AA0135, AA0576).

We thank the reviewer for pointing that out. We had also noticed these cells previously and chose not to cite the related manuscript. This is because the manuscript incorrectly states that there are no cells that project to both ET and IT targets, despite the presence of contradictory data in their data set. We've added a citation to the online data and make note of these cells in the results section.

Referee #2 (Remarks to the Author):

Nature manuscript 2020-03-05043

Zhang et al use a combination of snmC-seq and retrograde labeling of neuronal projections (Epi-Retro-Seq) to characterize the DNA methylation signatures of cortical projection neurons from 8 cortical areas and 10 target regions. The authors provide the rationale that high-resolution analysis of cortical IT and L5-ET neurons has not been possible based on previous approaches since these cell types are so closely related, and that Epi-Retro-Seq may help solve this issue.

The first 2 figures characterize the relationship between DNA methylation and source region, cortical layer, and target region. The authors observe that cortical layer is the primary determinant of methylation variability, followed by source region, then followed by target region. Many of the other analyses are expected and not very informative;

e.g. some cell types are more closely related to each other than other types. The authors identify differentially methylated genes (DMGs) between neurons of different projection targets, and GO analysis shows that these genes are enriched in neuronal and synaptic pathways, which is unsurprising.

In figures 3 and 4, the authors analyze the L5-ET neurons (the most abundant cell type in their dataset) more closely, and reach similar conclusions as above: source region plays a bigger role in determining methylation differences than target region, and DMGs between subclusters are enriched in canonical neuronal functions. Gene regulatory network analysis predicted key transcription factors that might underlie these DMGs, based on the presence of TF motifs in regions with differential CG methylation that are linked to particular target genes. However, without any experimental validation of this approach, it is unclear whether anything new is learned.

Finally, in figure 5 the authors validate the finding in their Epi-Retro-Seq data that a subset of L5-ET neurons project to both IT and ET targets, which is the main biological finding in the paper.

This work is of high technical quality and represents largely an in-depth characterization of the epigenetic relationships between different cortical projection neurons. However, it is unclear what biological insights to our understanding of brain physiology are gained from these analyses. A notable biological insight is the identification of L5-ET neuron subtypes that have both inter-telencephalic (IT) and extra-telencephalic (ET) projections, but the functional implications are not further explored. Thus, because the scientific question being investigated, though of substantial significance to neuroscientists, is quite narrow and may be of lesser interest to a wider scientific readership. I wonder, therefore, whether this work is of sufficient general interest for the broader readership of Nature rather than for a more focused journal. This, however, is an editorial decision.

We thank the reviewer for the positive comments on the comprehensiveness of the study and for recognizing that the work is “of substantial significance to neuroscientists”

We feel that there are several aspects of our findings that are of broad interest. We consider Neuroscience to be a broad field and note that Nature often publishes manuscripts with findings that are of more interest within disciplines such as immunology or cancer than neuroscience. Here we reiterate some of the broad implications of our findings for the field of neuroscience. In addition to the finding that there are L5-ET+CC cells, there are other findings that are of general importance and these are highlighted in each of the figures that we have presented. Very few previous studies have directly assessed the relationships between transcriptomics or epigenetics and cortico-cortical projections. We have provided data for a large number of projection source/target combinations and have shown: 1) that neurons projecting to different cortical targets do not form distinct clusters; 2) neurons projecting to different targets do vary statistically in their gene methylation; 3) the degree of similarity/difference varies depending on the particular sources and targets. Only one previous study (from our own work and now published in Neuron) has compared cortical neurons projecting to different cortical targets and this work selectively focused on comparing projections from a single area (VISp) to two other visual areas that are known to be the most distinct from each other and to have the most dedicated projections (Kim *et al.*, 2020). Thus, that study is likely to represent an outlying case and not necessarily representative of what we find with our much more comprehensive analyses.

MAJOR COMMENTS:

1) In terms of approach and statistical analyses, I don't have any major concerns, although it is difficult to assess some of the computational methods without having significant personal expertise in applying these programs. The biggest concern with statistics (mentioned below) is that it appears that only ~37 neurons on average are assayed per sample, which seems very underpowered.

We appreciate that sample size is an important determinant of the statistical power of the comparisons in our manuscript and thank the reviewer for the opportunity to clarify the sizes of our samples. We hope that the responses here and below will make it apparent that we have adequate sample sizes and will provide the reviewer with the resources required to assess sample sizes.

Based on this comment and a more detailed comment below, we infer that the reviewer has calculated the numbers of neurons sampled per source region per mouse. We note that samples from 2 mice were always pooled before FANS sorting and then the samples from 2 such replicates (male and female) were either pooled or considered independently depending on the analysis. Therefore, for each

projection/target combination, we actually sampled on average 168 cells (please see Extended Data Fig. 3a for details). Furthermore, all of the cells in our sample are considered in the context of a much larger “unbiased sample” that allowed them to be assigned to clusters. All of our statistical analyses have taken all this information into account. We are therefore confident that cells were correctly assigned to clusters and that the results accurately describe the relationships between methylation of single cells and their long distance projections.

2) Many of the figures/figure panels are unnecessarily complicated and confusing and don't always convey useful information. For example, in Figure 1D, it is almost impossible to get a sense of how much clustering there is by target region - could the same plot be color coded with just two colors - one for cortical and one for subcortical targets?

We apologize for any confusion. Here we explain the logic behind the organization and format of our figures. The point that we are attempting to make in Fig. 1d is that methylation of cells grouped according to target is not as distinct as according to source area or cluster identity. It is because the relationship of clustering to target regions is weak that the relationship is not readily apparent on the t-SNE plot. This contrasts with panels b and c where the relationships to clusters and source are clear.

We structured Fig. 1 to show essentially the same data in two different formats. The histograms in Fig. 1j provide the information that is not apparent in Fig. 1d. The t-SNE plots make the point that clustering according to targets is not as distinct as clustering according to sources, while the histograms are better suited to illustrating the relationships between clusters and projection targets.

We feel that the distinction between cells with cortical targets (IT cells) versus subcortical targets (L5-ET + L6-CT) is already apparent in Figs 1b-d. Fig 1b shows the locations in the t-SNE plot for the clusters of cells projecting to subcortical targets (L5-ET + L6-CT). It can also be seen from the color coding in Fig. 1d that the cells in the L5-ET cluster project to very different targets (have different colors) than all the other cells that are in other clusters.

In addition, Extended Data Fig 3a is far too complicated and does not convey much useful information that is not already presented in other figure panels and/or discussed in the text. The authors present every conceivable comparison between source region, target region, and layer, but the vast majority of these comparisons do not seem informative.

We agree that this information is not all necessary, but we do think that it is important and will be of interest to some readers. For example, we referred to this figure to

respond to the reviewer's question (above). The figure provides information that is not available elsewhere, such as the sample sizes for each experiment. And the individual Extended Data Fig. 3a panels show the breakdown of cells in each cluster for each individual source, rather than pooling them, as in Fig. 1j. This allows the reader to see the extent of enrichment of on-target cells for each source and projection target. We feel that this is appropriate as extended data, with the more condensed presentation reserved for the main figure.

3) The authors need to clarify the number of cells that were taken into account in their analysis per individual sample. It appears from the article that the entire study includes 320 samples for a total of 11,827 neurons analyzed. If this is the case, only an average of 37 neurons was analyzed per sample which could alter the clustering efficiency.

Please see response to 1) above. The numbers requested are shown in Extended Data Fig. 3a.

4) In Extended Data Fig.9, the authors compare Epi-Retro-Seq data with previously published Retro-seq data (scRNA-seq). This should be expanded to as much of the snmC-seq data set as possible to compare methylation with expression

We thank the reviewer for this suggestion which prompted us to look more carefully at the Retro-seq data (Tasic *et al.*, 2018). Both our study and the Retro-seq study sample from a limited number of the many possible combinations of sources and targets. The Retro-seq study included only ALM and VISp sources. The VISp source overlaps with one of our sources, but our studies differ in the targets sampled. For cortical targets, there is an overlap only for ACA. For subcerebral targets, there is overlap with SC, pons and thalamus, but it is clear in their data that the pons and SC are heavily contaminated with IT cells due to the artifact that we note in responses above and mitigate in our analyses using new exclusion criteria. (Note that our pons and SC target data are not contaminated because our injection pipette trajectories avoided paths through cortical areas.) In view of the very limited meaningful comparisons that would be possible, we have focused on the one comparison that is relevant to our main conclusion - medulla projecting cells are most distinct amongst the L5-ET cells.

5) Why are biological replicates not more related than by chance in Figure 1F? If replicates correspond to the same source regions from different animals, shouldn't they be correlated, if not as highly as neurons from the same source region in an individual animal?

We assume that the reviewer is referring to Fig. 1h since 1f is not related to replicates. The reviewer is correct that replicates should be correlated and this is

what is shown in the figure. The neighbor enrichment score is quantifying the local distinguishability of cells in the unsupervised embedding. Thus higher scores represent the groups that are more likely to form their own clusters, while lower scores mean that the groups are more intermingled or correlated. Thus the ~0.5 score of replicates means the different replicates are highly correlated. In comparison, neurons from different clusters, sources and targets are less correlated. To avoid the possibility that other readers will also find this to be unclear, we revised the sentence pasted below by adding the text that is highlighted in red (Line 147-149).

“Scores were near chance (neighbor enrichment score 0.5) for biological replicates, indicating that mCH profiles of different replicates are highly consistent (Fig. 1h).”

MINOR COMMENTS

1) The construction of the Epi-Retro-Seq vector is not described in the paper (and not referenced as all references to the method are to previous papers of the authors on methylation analysis). This makes it difficult and cumbersome to follow the approach. As I understand the very superficially described approach as outlined in figure 1a: They inject the Cre virus into the “target” region and retrogradely label the neurons that innervate the target. Retrogradely would mean that the virus has to travel via the axon back to the corresponding cell body in the source region as indicated in the figure. The cell bodies in the source would become GFP positive if Cre gets into the nucleus and thus these neurons can be isolated indicating as having been innervated the target region (the target of these neurons). Is my understanding of the approach correct? If so, how efficient is the infection of axons and labeling of cell bodies? Also: they need to explain the rationale for transduction of cholera toxin subunit B / Alexa red. The approach needs to be better explained to facilitate understanding.

The reviewer understands the mechanism of Epi-Retro-Seq correctly. We thank the reviewer for pointing out the need to clarify these details. To make descriptions of retrograde tracing more clear we have added a definition of retrograde tracing in the first paragraph of the results- “uptake of material injected into a target brain region and transported along axons, back to cell bodies at the source regions”. Both AAVretro and CTB are retrograde tracers.

We have also added a citation to Tervo et al. (Tervo *et al.*, 2016), which provides the original description and characterization of AAVRetro. The previous citation to Mo et al. (Mo *et al.*, 2015) includes not only the methylation analysis but also the design of the INTACT mouse line, where a Cre-dependent GFP was used to allow purification of genetically labeled cell-types.

The Tervo et al manuscript directly compares the efficiency of uptake of CTB and AAVRetro, but it should be noted that the efficiency of uptake does not influence our results. We use AAVretro to selectively sample a population of neurons projecting to a particular target but it would not matter if we labeled all such cells or only sampled a subset of them.

2) Is there concordance between differentially methylated genes (based on gene body CH methylation) and differential CG methylation at the promoters of these genes? That analysis would provide a nice snapshot of how concordant these two inferences of transcriptional activity are

Such correspondence has been extensively assessed in many previously published studies (Mo *et al.*, 2015; Luo *et al.*, 2017, 2019; Yao *et al.*, 2020a). Generally, both gene body mCH and mCG are anti-correlated with gene expression in neurons, while the correlation between promoter mCH and mCG are much weaker than gene body, since epigenetic features at promoter regions are usually less dynamic across cell-types (Roadmap Epigenomics Consortium *et al.*, 2015; Schultz *et al.*, 2015). Here, we performed two analyses to further validate that. First, we used L5-ET cells in our Epi- Retro-Seq data to test the correlation between gene body mCH and gene body mCG, promoter mCG across subclusters respectively, and observed a strong correlation between gene body mCH and mCG but not promoter mCG (Fig. R6a). Secondly, to evaluate the correlation of mCG and mCH with gene expression, we used our snmC2T- Seq data (Luo *et al.*, 2019), where DNA methylation and RNA expression were quantified in the same single nuclei simultaneously. This analysis also revealed the feature that has the strongest correlation with gene expression is mCH at gene-body (Fig. R6b).

We agree with the reviewer that the question of how well gene expression can be predicted from CG methylation is a very important one, but our data do not provide any new insight to these issues that cannot be obtained from other data sets, and we feel that it is beyond the scope of this manuscript to report on these expected relationships. Thus, we focus our descriptions and analyses on new insight that can be obtained by comparing neurons based on their projections, and therefore did not add this analysis to the manuscript.

Fig. R6. Correlation analysis between different data modalities. **a**, The Pearson correlation coefficients of mCH measured at gene body with mCG measured at different genomic regions. The correlations were computed across 15 L5-ET subclusters for 2,675 differentially methylated genes. **b**, The Pearson correlation coefficients of gene expression with mCH (blue) or mCG (orange) measured at different genomic regions. The correlations were computed across 17 major cell types of human prefrontal cortex for 2,154 differentially methylated genes ($FDR < 1e-10$, Kruskal-Wallis test).

3) Since FANS is known to induce epigenetic artifacts, such as drastic upregulation of immediate early genes (Fos, Jun, etc), and this may involve artifactual changes in DNA methylation, are there any single-cell methylation data on unsorted mouse cortex that can be used to quantify the extent of such artifacts in the Epi-Retro-Seq data?

Although manipulation of neurons can result in transcriptional changes in immediate early genes, we do not know of any evidence that FANS produces “epigenetic” artifacts. Further, published studies have shown under the conditions that we used, changes in the expression of immediate early genes is minimal (Lacar *et al.*, 2016). Any changes in gene methylation would be even less or non-existent over the very limited time between the preparation of nuclei and sorting before nuclei are frozen and then sequenced. This was already cited in our Methods section.

4) It would be more appropriate to use the phrase “DNA methylation diversity” and “DNA methylation differences” rather than “Epigenetic diversity” and “Epigenetic differences”

We agree that this might be a more specific term that could be used, however “epigenetic” is accurate and encompasses DNA methylation. We have used epigenetic in the places where the alternatives are more cumbersome. We expect that readers will not be confused by this because we make clear that all of the data are based on assessment of DNA methylation.

5) The resolution of Figure 2 should be improved.

This has been corrected.

6) Results should be described in the same order as they appear in the figures to facilitate the reading.

We have arranged the panels in our figures to minimize the amount of wasted space. Accordingly, we also label the figure panels according to their organization within the figure. If the figure panels were re-labeled according to the order in which we present the data, then the organization of the figures would be jumbled. We have done our best to minimize this problem and have reviewed the text to assure that readers will not be confused. If the editor prefers, we could rearrange the figure at the expense of creating wasted white space within the figures.

7) Line 1087, replace “Epi-seq” by “Retro-seq”.

The word has been corrected.

8) In line 390, replace “expression” with “methylation” The word has been corrected.

References

- 1) Kim, E. J. *et al.* (2020) ‘Extraction of Distinct Neuronal Cell Types from within a Genetically Continuous Population’, *Neuron*. doi: 10.1016/j.neuron.2020.04.018.
- 2) Lacar, B. *et al.* (2016) ‘Nuclear RNA-seq of single neurons reveals molecular signatures of activation’, *Nature communications*. nature.com, 7, p. 11022.
- 3) Liu, H. *et al.* (2020) ‘DNA Methylation Atlas of the Mouse Brain at Single-Cell Resolution’, *bioRxiv*. doi: 10.1101/2020.04.30.069377.
- 4) Luo, C. *et al.* (2017) ‘Single-cell methylomes identify neuronal subtypes and regulatory elements in mammalian cortex’, *Science*, 357(6351), pp. 600–604.
- 5) Luo, C. *et al.* (2019) ‘Single nucleus multi-omics links human cortical cell regulatory genome diversity to disease risk variants’, *bioRxiv*. doi: 10.1101/2019.12.11.873398.
- 6) Mo, A. *et al.* (2015) ‘Epigenomic Signatures of Neuronal Diversity in the Mammalian Brain’, *Neuron*, 86(6), pp. 1369–1384.
- 7) Oberlaender, M. *et al.* (2012) ‘Cell type-specific three-dimensional structure of thalamocortical circuits in a column of rat vibrissal cortex’, *Cerebral cortex* ,

22(10), pp. 2375–2391.

- 8) Roadmap Epigenomics Consortium *et al.* (2015) 'Integrative analysis of 111 reference human epigenomes', *Nature*, 518(7539), pp. 317–330.
- 9) Schultz, M. D. *et al.* (2015) 'Human body epigenome maps reveal noncanonical DNA methylation variation', *Nature*, 523(7559), pp. 212–216.
- 10) Tasic, B. *et al.* (2016) 'Adult mouse cortical cell taxonomy revealed by single cell transcriptomics', *Nature neuroscience*, 19(2), pp. 335–346.
- 11) Tasic, B. *et al.* (2018) 'Shared and distinct transcriptomic cell types across neocortical areas', *Nature*, 563(7729), pp. 72–78.
- 12) Tervo, D. G. R. *et al.* (2016) 'A Designer AAV Variant Permits Efficient Retrograde Access to Projection Neurons', *Neuron*. Elsevier, 92(2), pp. 372–382.
- 13) Yao, Z. *et al.* (2020a) 'An integrated transcriptomic and epigenomic atlas of mouse primary motor cortex cell types', *Biorxiv*. biorxiv.org. Available at: <https://www.biorxiv.org/content/10.1101/2020.02.29.970558v1.abstract>.
- 14) Yao, Z. *et al.* (2020b) 'A taxonomy of transcriptomic cell types across the isocortex and hippocampal formation', *Biorxiv*. biorxiv.org. Available at: <https://www.biorxiv.org/content/10.1101/2020.03.30.015214v1>.

Reviewer Reports on the First Revision:

Referees' comments:

Referee #1 (Remarks to the Author):

These comments on the revised MS are in addition to the original review comments:

The revised manuscript largely addresses this review's previous comments, questions, and concerns, and it improved substantially in the revision process. Overall, the integrated analysis of neuronal connectivity and epigenomic features at single-cell resolution is a novel and promising direction of research. The authors employ a range of computational regression analysis tools to illuminate potential relationships between cortical projection neuron types, their axonal targets, and methylome features. The study corroborates very considerable previous molecular and anatomical knowledge on projection neuron diversity in the cerebral cortex, and provides new experimental approaches and resources that help disentangle epigenetic/molecular regulations underlying cortical projection neuron diversity and connectivity.

In the original submission, the data presented by the authors indicated the presence of a considerable degree of "contaminant" (off-target) cells through suboptimal or erroneous injections/FANS. In the revised manuscript, the authors put substantial effort into dissecting out and eliminating such suboptimal experiments and data.

By comparing the observed versus expected proportions of different types of neurons for a select set of source-target pairs (based on their other study in preprint; Liu *et al.*, BioRxiv, 2020), the authors assign a subjective threshold value for the maximum tolerable degree of contamination for ET (extra-telencephalic) and IT (intra-telencephalic) samples. They then apply this threshold with

the aim to eliminate samples above the threshold contamination. Using these new exclusion criteria, they decided to eliminate 36 suboptimal experiments (out of 216; ~17%), with one additional experiment eliminated due to an artifact caused by FANS-gating of small sample size. The authors find that the main conclusions of the study mostly remained unchanged.

As the authors are aware, these newly applied exclusion criteria are biased toward eliminating only the type of suboptimal experiments that involve target areas with a clear difference between IT vs. PT cell-type proportions. Other non-excluded experiments could still be quite suboptimal in this regard, simply not detected by these "high-contrast" exclusion criteria. Indeed, the six eliminated ET samples turned out to be the ones that contain a high percentage of IT cells seemingly projecting to the thalamus or ventral tegmental area, and this is likely because there should be few IT cells, if any, projecting to the TH or VTA from cortical source areas. The authors attribute such contamination to the mislabeling of source cells by the retrograde injection needle passing through the source areas in the cortex. Similarly, the 30 suboptimal IT samples were identified based on the number of layer VI corticothalamic cells (L6-CT) or interneurons present in the sample; e.g., if a sample contained a high percentage L6-CT that seemingly project to the striatum, that sample would be eliminated, because there is a consensus that L6-CT cells do not project to the striatum. This entire enterprise still appears highly problematic, both for taxonomic science and for discovery biology. The interpretations are circularly based on expectations and assume uniformity of connectivity by all neurons with somewhat over-simplified, "recognizable", nameable type.

These exclusion criteria are based on somewhat simplified expectations, and are somewhat arbitrary, so they likely do not work so well for all target areas (such as the striatum, where all PT axons pass through, and many IT cells have collaterals). In spite of such selection biases, this simplified, practical approach taken by the authors is reasonable, logical, and is the best they were able to do with the experiments they had. Further effort does not seem justified to more accurately determine other sources of off-target cells in the samples, since it would potentially have a small impact on their main conclusions. However, the limitations should be pointed out and discussed clearly in the text and figures so readers do not think that this somewhat simplified assessment is definitive, or that it encompasses the full depth of nuance and diversity in the system. It would be especially useful to discuss the 37 excluded experiments clearly and explicitly, including how and if they might have altered some interpretations, to aid others thinking of follow-on work.

Typos:

Line 820: On- and Off-target symbols should be switched.

Line 494: the words "sources" and "areas" are redundant.

Referee #2 (Remarks to the Author):

In the revision of their manuscript, Zang et al significantly improved the overall quality of the text and figures. Although the latter could be further simplified, the revised version adds more details which clarify the message. The authors also commented properly on the link between differentially methylated genes and gene expression. In this form, the study is well elaborated, easier to understand, and suitable for publication.

The authors commented on the significance of this study for the broad audience of Nature. In my opinion a slight concern remains whether the study is of sufficient general biological interest for readers of Nature or may be more appropriate for a specialized audience. This is, however, an editorial decision.